# Tissue-specific regulation of BMP signaling by *Drosophila N*-glycanase 1

**Antonio Galeone[1], Seung Yeop Han[1], Chengcheng Huang[2], Akira Hosomi[2], Tadashi Suzuki[2], Hamed Jafar-Nejad[1,3]***

[1]Department of Molecular and Human Genetics, Baylor College of Medicine, Houston, United States; [2]Glycometabolome Team, RIKEN Global Research Cluster, Saitama, Japan; [3]Program in Developmental Biology, Baylor College of Medicine, Houston, United States

**Abstract** Mutations in the human *N*-glycanase 1 (*NGLY1*) cause a rare, multisystem congenital disorder with global developmental delay. However, the mechanisms by which *NGLY1* and its homologs regulate embryonic development are not known. Here we show that *Drosophila Pngl* encodes an *N*-glycanase and exhibits a high degree of functional conservation with human NGLY1. Loss of *Pngl* results in developmental midgut defects reminiscent of midgut-specific loss of BMP signaling. *Pngl* mutant larvae also exhibit a severe midgut clearance defect, which cannot be fully explained by impaired BMP signaling. Genetic experiments indicate that Pngl is primarily required in the mesoderm during *Drosophila* development. Loss of Pngl results in a severe decrease in the level of Dpp homodimers and abolishes BMP autoregulation in the visceral mesoderm mediated by Dpp and Tkv homodimers. Thus, our studies uncover a novel mechanism for the tissue-specific regulation of an evolutionarily conserved signaling pathway by an *N*-glycanase enzyme.

DOI: https://doi.org/10.7554/eLife.27612.001

***For correspondence:**
hamedj@bcm.edu

**Competing interests:** The authors declare that no competing interests exist.

## Introduction

*NGLY1* (*N*-glycanase 1) encodes an evolutionarily conserved enzyme that catalyzes the cleavage of *N*-glycans from glycoproteins (*Suzuki et al., 2000*). Whole-genome and -exome sequencing has recently resulted in the identification of *NGLY1* mutations in patients with an autosomal recessive developmental disorder called NGLY1 deficiency (*Caglayan et al., 2015*; *Enns et al., 2014*; *Heeley and Shinawi, 2015*; *Need et al., 2012*). NGLY1-deficient patients show a host of phenotypes including global developmental delay, movement disorder, hypotonia, absent tears, peripheral neuropathy, constipation, and small feet and hands (*Enns et al., 2014*; *Lam et al., 2017*). The mechanism by which NGLY1 deficiency causes the above-mentioned clinical phenotypes is not known, and neither has NGLY1 been linked to any major developmental signaling pathway.

In yeast, *N*-glycanase 1 (Peptide: *N*-glycanase or PNGase, encoded by *PNG1*) has been associated with endoplasmic reticulum-associated degradation (ERAD) (*Suzuki, 2007*; *Suzuki et al., 2016*), a process that plays a crucial role in the proteasome-mediated degradation of misfolded proteins (*Brodsky, 2012*; *Smith et al., 2011*). However, null mutants for yeast *PNG1* do not show apparent phenotypic abnormalities and exhibit normal growth rate and viability under a variety of experimental conditions (*Suzuki et al., 2000*). A recent study has provided strong evidence that in *C. elegans*, a transcription factor called SKN-1 (homolog to mammalian NFE2L1/2) needs to be retrotranslocated from ER to cytoplasm by ERAD machinery and deglycosylated by NGLY1 (PNG-1 in worms) to function properly (*Lehrbach and Ruvkun, 2016*). However, deglycosylation does not result in SKN-1 degradation, but instead promotes the activation of SKN-1 so that it can mediate the necessary transcriptional response to proteasome disruption (*Lehrbach and Ruvkun, 2016*). Moreover, mouse embryonic fibroblasts lacking NGLY1 did not show impairment and/or delay in the

**eLife digest** DNA carries the information needed to build and maintain an organism, and units of DNA known as genes contain coded instructions to build other molecules, including enzymes. Sometimes, genes can become faulty and develop mutations that can affect how an embryo develops and lead to diseases. For example, people with mutations in the gene that encodes an enzyme called *N*-glycanase 1 experience many problems with their nervous system, gut and other organs.

Normally, *N*-glycanase 1 helps the body remove specific sugar molecules from some proteins in the cells, and is also thought to be important during embryonic development. As an embryo develops, its cells undergo a series of transformations, which is regulated by different molecules and signaling pathways. For example, a pathway known as BMP signaling plays an important role in many tissues. Problems with this pathway can lead to many diseases throughout the body, including the gut, where it helps cells to develop.

Previous research has shown that fruit flies lacking the gene that codes for an equivalent *N*-glycanase enzyme (which is called Pngl in flies) cannot develop properly into adults. However, until now it was not known what type of cells need the *N*-glycanase enzyme in any organism, or if NGLY1 is essential for important signaling pathways like BMP signaling. Now, Galeone et al. have used genetically modified flies to test how losing Pngl affected their development.

The results first showed that engineering Pngl-deficient fruit flies to produce the human enzyme eliminated their problems; these flies developed and survived like normal flies. This confirmed that that the human and fly enzymes can perform equivalent roles. Galeone et al. then discovered that Pngl plays two distinct roles in a group of cells that surround the fruit fly's gut tissue and give rise to the cells that eventually form the muscle layer in the gut. In the larvae, Pngl was required to empty the gut, which is a necessary step before the larvae can develop into an adult. Moreover, Pngl is needed for BMP signaling in the gut, and when flies had the enzyme removed, some parts of their gut could not from properly.

This study will provide a framework to improve our understanding of how BMP signaling is regulated in humans. A next step will be to test if some of the symptoms experienced by patients without a working copy of the gene for *N*-glycanase 1 are caused by a faulty BMP-signaling system in specific tissues. If this is the case, it could provide new opportunities to treat some of these symptoms.

DOI: https://doi.org/10.7554/eLife.27612.002

degradation of misfolded proteins, but displayed an unconventional deglycosylation reaction that may generate aggregation-prone proteins harboring *N*-GlcNAc and result in cell toxicity in the absence of NGLY1 (*Huang et al., 2015*). Therefore, although NGLY1 seems to be functionally associated with the ERAD machinery, it does not seem to directly promote the degradation of misfolded proteins in animals.

Human *NGLY1* has a single *Drosophila* homolog called PNGase-like (*Pngl*), whose loss-of-function mutants result in semi-lethality and sterility (*Funakoshi et al., 2010*). We have used *Drosophila* to determine the molecular mechanisms by which *Pngl* regulates animal development. Our data indicate that *Pngl* is required in the visceral mesoderm (VM) during midgut development to regulate bone morphogenetic protein (BMP) signaling from VM to endoderm in embryonic midgut and also to ensure midgut clearance before puparium formation. BMP ligands can signal either as homodimers or as heterodimers through homodimeric or heterodimeric BMP type I receptors (*Bangi and Wharton, 2006a*; *Ray and Wharton, 2001*). Our data indicate that loss of *Pngl* abolishes a BMP autoregulatory loop in the VM mediated by homodimers of one ligand (Decapentaplegic; Dpp) acting through homodimers of one receptor (Thickveins; Tkv), and hence link a deglycosylation enzyme to tissue-specific regulation of BMP signaling in flies.

## Results

### A high level of functional conservation exists between fly Pngl and human NGLY1

Based on the analysis of three *Pngl* alleles (*Pngl^{ex14}*, *Pngl^{ex18}* and *Pngl^{ex20}*; *Figure 1A*), it has previously been reported that loss of *Drosophila Pngl* results in developmental delay and a semi-lethality phenotype, with about 1% adult escapers (*Funakoshi et al., 2010*). Moreover, these phenotypes were rescued by ubiquitous expression of mouse NGLY1 (*Funakoshi et al., 2010*), suggesting functional conservation between Pngl and its mouse homolog. To further examine the degree of conservation between fly Pngl and its mammalian homologs, we used ΦC31-mediated transgenesis (*Bischof et al., 2007*; *Venken et al., 2006*) to generate transgenic flies capable of overexpressing wild-type (WT) human NGLY1 or the NGLY1-ΔR402 mutant, a single amino acid in-frame deletion identified in an NGLY1 deficiency patient (*Enns et al., 2014*), and asked whether they can rescue the homozygous lethality of *Pngl^{ex14}* and *Pngl^{ex18}*. Ubiquitous expression of WT human NGLY1, but not NGLY1-ΔR402, was able to rescue the lethality in both *Pngl* alleles (*Figure 1B*). Both male and female *Pngl^{−/−}* escaper flies are sterile and short-lived (*Figure 1C and D*) (*Funakoshi et al., 2010*). However, adult *Pngl^{−/−}; Act > NGLY1* WT animals do not show these phenotypes (*Figure 1C and D*). Together, these results underscore the *in vivo* functional homology between *Pngl* and *NGLY1*.

A previous study did not detect PNGase activity in wild-type *Drosophila* larval extracts in an *in vitro* assay using $^{14}$C-labeled asialofetuin glycopeptide as a substrate and concluded that *Drosophila* Pngl might not possess *N*-glycanase activity (*Funakoshi et al., 2010*). To better assess whether Pngl can function as an *N*-glycanase, we used RTL (RTAΔ-transmembrane-Leu2) spotting assays, which provide a reproducible *in vivo* model to assess the level of PNGase activity in yeast (*Hosomi et al., 2010*). RTL undergoes ERAD in a PNGase-mediated, deglycosylation-dependent manner and therefore leucine-auxotrophic yeast cells which express functional *Saccharomyces cerevisiae* PNGase (Sc-Png1) are unable to grow in media lacking leucine. However, yeast cells that lack Sc-Png1 or express a catalytically-inactive version of Sc-Png1 fail to degrade RTL and can therefore grow on leucine-deficient medium (*Hosomi et al., 2010*). As shown in *Figure 1E*, *png1* mutant yeast cells (*png1Δ*) grow well in media with or without leucine when transfected with the RTL plasmid. Expression of Sc-Png1-HA severely decreases the ability of these cells to grow in the absence of leucine, confirming that this phenotype is PNGase-dependent (*Figure 1E*) (*Hosomi et al., 2010*). An HA-tagged version of the fly Pngl suppressed the growth of the yeast cell in media without leucine to the same extent as the Sc-Png1 (*Figure 1E*). Meanwhile, Pngl harboring a C303A mutation in its putative catalytic domain (Pngl-HA-C303A) failed to rescue PNGase function (*Figure 1E*), even though WT and C303A versions are expressed at comparable levels (*Figure 1F*). These observations strongly suggest that *Drosophila* Pngl is able to deglycosylate RTL and facilitate the efficient degradation of the RTL protein. Next, we expressed a FLAG-tagged version of RTAΔ in yeast cells and asked whether Pngl can increase the level of the deglycosylated version of this protein upon inhibition of the protein biosynthesis. In *png1Δ* mutant yeast cells, almost all RTAΔ was found to be in the glycosylated form (g1) at three time points (*Figure 1G*). Expression of Sc-Png1 or fly Pngl increased the relative level of the deglycosylated form (g0) in a time-dependent manner (*Figure 1G and H*). However, Pngl-C303A failed to remove glycans from RTAΔ. Similar experiments indicate that human NGLY1, but not the C309A catalytic mutant NGLY1, can also compensate for the lack of Sc-Png1 in the RTL assay (*Figure 1I*). Of note, NGLY1-ΔR402 showed a weak rescue of the PNGase activity in this assay, suggesting that it is a hypomorphic allele. Altogether, these observations indicate that *Drosophila* Pngl is indeed an *N*-glycanase enzyme and that there is high level of functional conservation between Pngl and its yeast and human homologs.

### Loss of *Pngl* causes gastric caeca and acidification defects in the larval midgut

*Pngl^{−/−}* larvae did not show any gross morphological defects. However, inspection of the larval internal organs suggested defects in the mutant larval midguts. In the anterior part of the midgut, control larvae harbor four finger-like structures called gastric caeca (*Figure 2A and B* and *Figure 2—figure supplement 1*), which play key roles in the insect digestive system, including water and ion transport and secretion of enzymes (*Pullikuth et al., 2006*; *Volkmann and Peters, 1989*). *Pngl^{−/−}* larvae

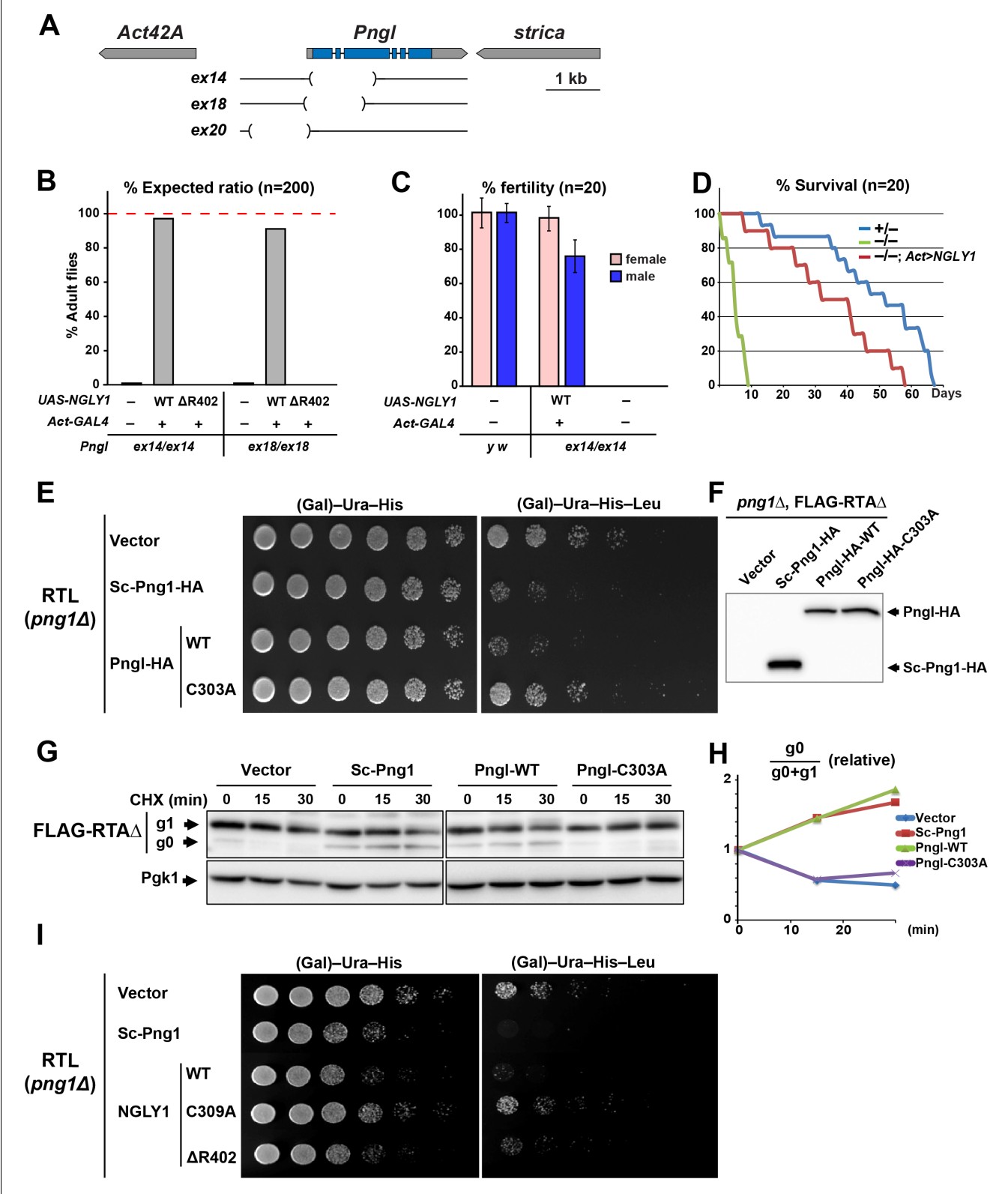

**Figure 1.** Fly Pngl has a high level of functional conservation with human NGLY1. (**A**) Schematic of the *Pngl* locus and the deleted section of the alleles used in this study. (**B**) Eclosion tests of *Pngl*[−/−] flies with or without rescue by ubiquitous expression of NGLY1-WT or NGLY1-ΔR402. The red dashed line marks the expected Mendelian ratio. (**C**) Fertility tests of *Pngl*[−/−] flies rescued by NGLY1-WT and *Pngl*[−/−] escaper flies compared to yellow white (*y w*) flies used as control. (**D**) Longevity tests of *Pngl*[−/−] escaper flies compared to *Pngl*[+/−] and *Pngl*[−/−] flies rescued by NGLY1-WT. (**E**) RTL spotting assay on

*Figure 1 continued on next page*

Figure 1 continued

png1Δ mutant yeast transfected with empty vector or expression vectors for HA-tagged versions of *Saccharomyces cerevisiae* PNGase (Sc-Png1-HA), and wild-type or C303A-mutant *Drosophila* Pngl (Pngl-HA). (F) Western blot analysis of the yeast strains used in (E) by anti-HA antibody. (G) Cycloheximide (CHX) decay assay for FLAG-RTAΔ on yeast transfected with the same vectors as (E), followed by immunoblotting with anti-FLAG. Phosphoglycerate kinase (Pgk1) was used as loading control. (H) Quantification of cycloheximide decay assay for FLAG-RTAΔ showing deglycosylated (g0)/deglycosylated (g0) +glycosylated (g1) ratio for each genotype over time. The graph represents the mean of three independent experiments. (I) RTL spotting assay by using yeast PNGase (Sc-Png1-HA), and WT, C309A or ΔR402-mutant human NGLY1.
DOI: https://doi.org/10.7554/eLife.27612.003

showed a severe shortening of the gastric caeca (*Figure 2C*, red asterisks) compared to *y w* and heterozygote controls (*Figure 2B* and *Figure 2—figure supplement 1*). Control larvae harbor a specific region in the middle midgut called the 'acid zone', which has a luminal pH of less than 2.3 and is considered the fly stomach (*Figure 2A*) (*Dubreuil, 2004*). The acid zone can be visualized by feeding larvae with bromophenol blue (BPB), which is blue in neutral and basic pH but turns yellow as the pH is decreased from 7.0 to 1.0 (*Figure 2D* and *Figure 2—figure supplement 1*, red dotted box). *Pngl⁻/⁻* larvae are almost completely devoid of the yellow color in the midgut, indicating a loss of acid zone (*Figure 2E*). Similar to *Pngl^ex14/ex14* larvae, *Pngl^ex18/ex18* and *Pngl^ex20/ex20* animals showed shortened gastric caeca and a loss of acid zone (*Figure 2—figure supplement 2* and not shown). Moreover, *Pngl^ex14* and *Pngl^ex18* hemizygous animals (over a deficiency allele) showed semi-lethality and midgut phenotypes (*Figure 2—figure supplement 2* and not shown), suggesting that *ex14* and *ex18* deletions are genetic null alleles. Altogether, these data indicate that loss of *Pngl* leads to specific midgut defects in *Drosophila* larvae.

## BMP signaling from VM to endoderm is impaired upon loss of *Pngl*

In *Drosophila* embryos, a member of the bone morphogenetic protein (BMP) family called Decapentaplegic (Dpp) signals from VM to endoderm and is required for midgut specification (*Dubreuil, 2004*; *Panganiban et al., 1990*). Specifically, BMP signaling from VM in parasegment 3 (PS3) and PS7 is transduced by phosphorylated Mothers against dpp (pMad) (*Newfeld et al., 1996*) in midgut endoderm, resulting in the formation of the gastric caeca and the acid zone, respectively (*Figure 2F*). Given the striking similarity between *Pngl* midgut phenotypes and those caused by impaired BMP signaling from VM (*Panganiban et al., 1990*), we examined the effects of loss of *Pngl* on BMP signaling by staining *Drosophila* embryos with an antibody against human pSMAD3, which recognizes *Drosophila* pMad (*Li et al., 2016*). Control embryos (*y w* and *Pngl⁺/⁻*) showed pMad staining in areas corresponding to PS3 and PS7 (*Figure 2G and G'* and *Figure 2—figure supplement 1*). However, *Pngl⁻/⁻* embryos showed a dramatic decrease in the level of pMad in PS3 and PS7 (*Figure 2H and H'*). Notably, pMad staining in other regions of the embryos, including the ectodermal and head regions, was not affected by the loss of *Pngl* (*Figure 2H'*, compare to *Figure 2G'* and *Figure 2—figure supplement 1*). Loss of BMP signaling in PS7 results in lack of second midgut constriction and impairs the expression of the acid-secreting (copper) cell-specific homeodomain gene *labial* in PS7 endodermal cells (*Immerglück et al., 1990*; *Nellen et al., 1994*; *Panganiban et al., 1990*). We used anti-Labial antibody to mark the precursors of the midgut copper cells in the embryonic endoderm and anti-Fas3 (Fasciclin 3) antibody to mark VM (*Weiss et al., 2001*) and to visualize midgut constrictions and chambers. As reported previously (*Nellen et al., 1994*), at stage 15, control embryos show a prominent second midgut constriction and Labial expression anterior to it (*Figure 2I and I'* and *Figure 2—figure supplement 1*). However, in stage 15 *Pngl⁻/⁻* embryos, no Labial signal was detectable in the PS7 region (*Figure 2J and J'*). By stage 16, all three constrictions are clearly visible in control larvae and Labial is expressed in the midgut compartment between first and second constrictions (*Figure 2K and K'* and *Figure 2—figure supplement 1*). However, stage 16 *Pngl⁻/⁻* embryos lack Labial staining and second constriction (*Figure 2L and L'*), in agreement with impaired BMP signaling in PS7. Together, these data demonstrate that *Pngl* is required for mesoderm-to-endoderm BMP signaling in *Drosophila* embryos.

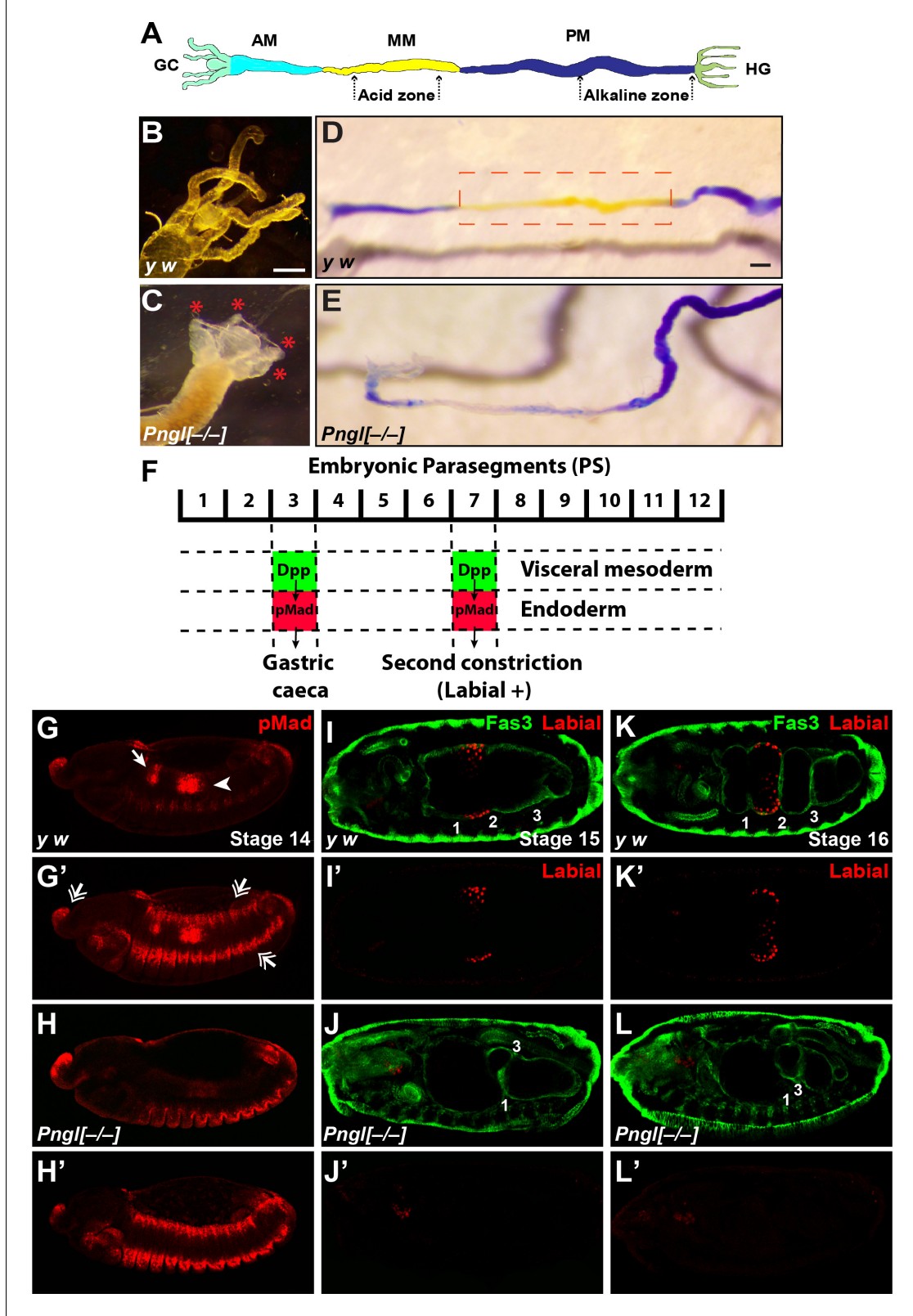

**Figure 2.** Loss of *Pngl* results in larval midgut defects and impaired BMP signaling from VM to endoderm. (**A**) Schematic drawing of larval midgut/ hindgut indicating gastric caeca (GC), anterior midgut (AM), middle midgut (MM), posterior midgut (PM), and hindgut (HG). (**B and C**) Bright field images of the proximal midgut region of larvae 96 hours after egg deposition. Red asterisks in C mark the shortened gastric caeca upon loss of *Pngl*. Scale bar, 100 µm. (**D and E**) Bright field images of midgut from third instar larvae fed with food containing bromophenol blue. Acid zone is delimited

*Figure 2 continued on next page*

*Figure 2 continued*
by the red dotted boxes. Scale bar, 100 μm. (F) Schematic drawing of Dpp signaling in the embryonic midgut. (G–H') pMad staining of stage 14 embryos of the indicated genotypes. In G and H, limited projection views are shown to highlight pMad expression in PS3 (arrow) and PS7 (arrowhead). G' and H' are full projection view of the datasets shown in G and H. Note that expression of pMad in ectodermal bands and other regions are not affected by the loss of *Pngl* (double-arrowheads). (I–L') Fas3 (VM marker) and Labial staining of stage 15 and 16 embryos of indicated genotypes. Midgut constrictions are marked by numbers along the anterior-posterior axis.
DOI: https://doi.org/10.7554/eLife.27612.004
The following figure supplements are available for figure 2:

**Figure supplement 1.** Loss of one copy of *Pngl* does not impair midgut development and BMP signaling in the embryo.
DOI: https://doi.org/10.7554/eLife.27612.005
**Figure supplement 2.** *Pngl^{ex14}* and *Pngl^{ex18}* are genetic null alleles.
DOI: https://doi.org/10.7554/eLife.27612.006

## *Pngl* is primarily required in the mesoderm during *Drosophila* development, but not all *Pngl* phenotypes can be explained by impaired Dpp signaling

To determine which tissues or cell types require the function of Pngl, we performed RNAi-mediated knock-down (KD) and rescue experiments. When crossed to the mesodermal driver *Mef2-GAL4*, the *Pngl^{RNAi}* KK101641 strain (http://stockcenter.vdrc.at/control/main) resulted in 100% lethality at room temperature and recapitulated the gastric caeca shortening and acid zone loss phenotypes observed in *Pngl* mutants (*Figure 3A–C*). In addition, *Pngl* KD by another mesodermal driver called *how^{24B}-GAL4* resulted in partial lethality and acid zone defects in larvae (*Figure 3A and E*). *Pngl* KD with *how^{24B}-GAL4* did not affect gastric caeca formation (*Figure 3D*), likely because this driver induces transgene expression later than *Mef2-GAL4* (*Figure 3—figure supplement 1*), after gastric caeca anlagen have already been formed. However, *Pngl* KD by two endodermal drivers did not result in lethality and gut phenotypes (*Figure 3A and F–I*). Moreover, overexpression of human *NGLY1* with *Mef2-GAL4* and *how^{24B}-GAL4* rescued the lethality of the *Pngl* alleles (*Figure 3J*). In agreement with the severity of their corresponding KD phenotypes and expression patterns, *Mef2-GAL4* rescued the lethality more efficiently than *how^{24B}-GAL4* (*Figure 3J*). Together, these data indicate that during *Drosophila* development Pngl is primarily required in the mesoderm to ensure animal survival and to promote BMP signaling from visceral mesoderm to endoderm.

It was previously shown that a *Pngl* transgene harboring the C303A mutation in the catalytic domain (*Figure 1E and G*) is not able to rescue the lethality of *Pngl* mutants (*Funakoshi et al., 2010*). To test whether the enzymatic activity of Pngl is required for the regulation of BMP signaling in the midgut, we overexpressed *Pngl-C303A* by *Mef2-GAL4* in *Pngl* mutants. As shown in *Figure 3K and L*, overexpression of wild-type *Pngl* rescues gastric caeca and acid zone defects of *Pngl* mutants. However, the mutant version fails to rescue *Pngl* midgut phenotypes (*Figure 3M and N*), even though Western blot shows that this mutation does not affect the expression level or stability of Pngl (*Figure 3O*). Together, these data indicate that the BMP signaling impairment observed in *Pngl* larval midguts is due to the lack of Pngl's enzymatic activity.

Previous reports indicate that in the absence of midgut acidification, larvae are able to reach adulthood (*Dubreuil et al., 2001*). Moreover, regulatory mutations that result in specific loss of *dpp* expression in PS3 and a complete loss of gastric caeca only show a partial lethality (*Masucci and Hoffmann, 1993*). Accordingly, impaired Dpp signaling in the midgut is not sufficient to fully explain the lethality of *Pngl* mutants, especially given that gastric caeca are not completely lost. Indeed, animals undergoing *dpp* KD by the same mesodermal drivers showed a higher survival compared to those undergoing *Pngl* KD (*Figure 3P*, compare to 3A). Specifically, *Mef2 > Pngl^{RNAi}* and *how^{24B} > Pngl^{RNAi}* animals showed 0% and ~28% survival rate into adulthood, but *Mef2 > dpp^{RNAi}* and *how^{24B} > dpp^{RNAi}* animals showed ~8% and ~80% survival rate into adulthood, respectively. The weaker effects of *dpp^{RNAi}* are not likely to result from inefficient *dpp* KD, as *Mef2 > dpp^{RNAi}* larvae exhibited a complete loss of gastric caeca and acid zone, and *how^{24B} > dpp^{RNAi}* larvae lost the acid zone but not the gastric caeca, suggesting a loss or severe impairment of mesoderm-to-endoderm BMP signaling corresponding to the spatiotemporal expression pattern of each driver (*Figure 3Q–T*). These observations indicate that impaired Dpp signaling in the midgut can only partially account for the lethality observed in *Pngl* mutants.

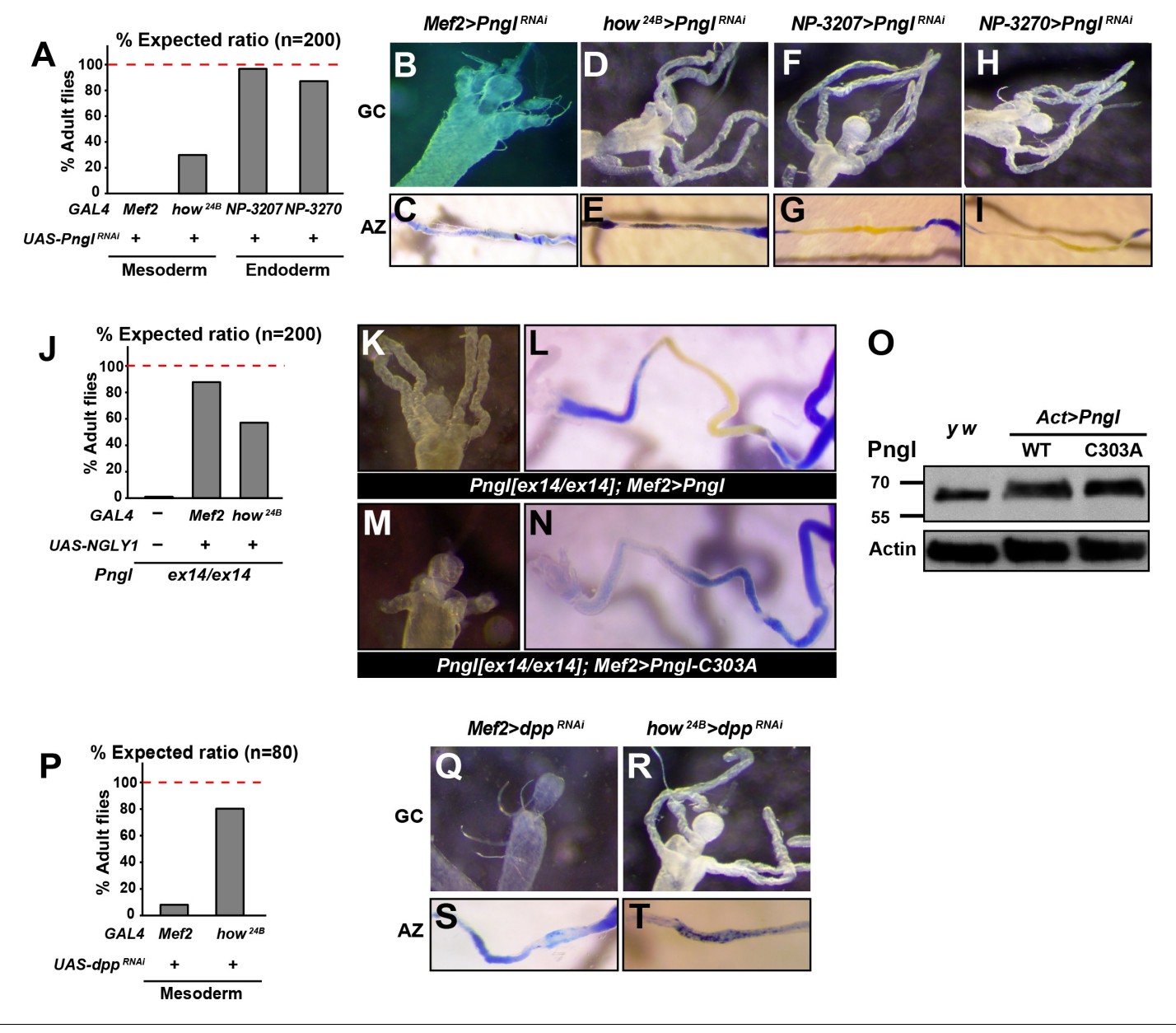

**Figure 3.** The enzymatic activity of Pngl is essential in the mesoderm for the regulation of BMP pathway in the midgut. (A) Eclosion tests of *Pngl*[RNAi] flies based on expected Mendelian ratio using two pan-mesodermal drivers (*Mef2-* and *how*[24B]*-GAL4*) and two midgut endodermal drivers (*NP3207-* and *NP3270-GAL4*). (B–I) Proximal midgut region and acid zone of the indicated genotypes. (J) Rescue of the lethality of *Pngl*[ex14/ex144] animals by expressing NGLY1-WT using *Mef2-* and *how*[24B]*-GAL4* drivers. (K–N) Proximal midgut region and acid zone of the indicated genotypes. (O) SDS gels were used to run larval extracts from the indicated genotypes and were probed with an antibody against Pngl. (P) Eclosion tests of *dpp*[RNAi] flies using *Mef2-* and *how*[24B]*-GAL4* drivers. (Q–T) Proximal midgut region and acid zone of the indicated genotypes.
DOI: https://doi.org/10.7554/eLife.27612.007

The following figure supplement is available for figure 3:

**Figure supplement 1.** *Mef2-GAL4* expression starts earlier than *how*[24B]*-GAL4* expression during embryonic development.
DOI: https://doi.org/10.7554/eLife.27612.008

So far our data suggest that Pngl also plays a Dpp-independent role in the mesoderm during *Drosophila* development. We noticed that during the wandering phase, *Pngl*[−/−] larvae failed to empty their guts (*Figure 4A and A'*). To better characterize this phenotype, we carried out midgut clearance assays. When larvae are raised on regular food supplemented with bromophenol blue

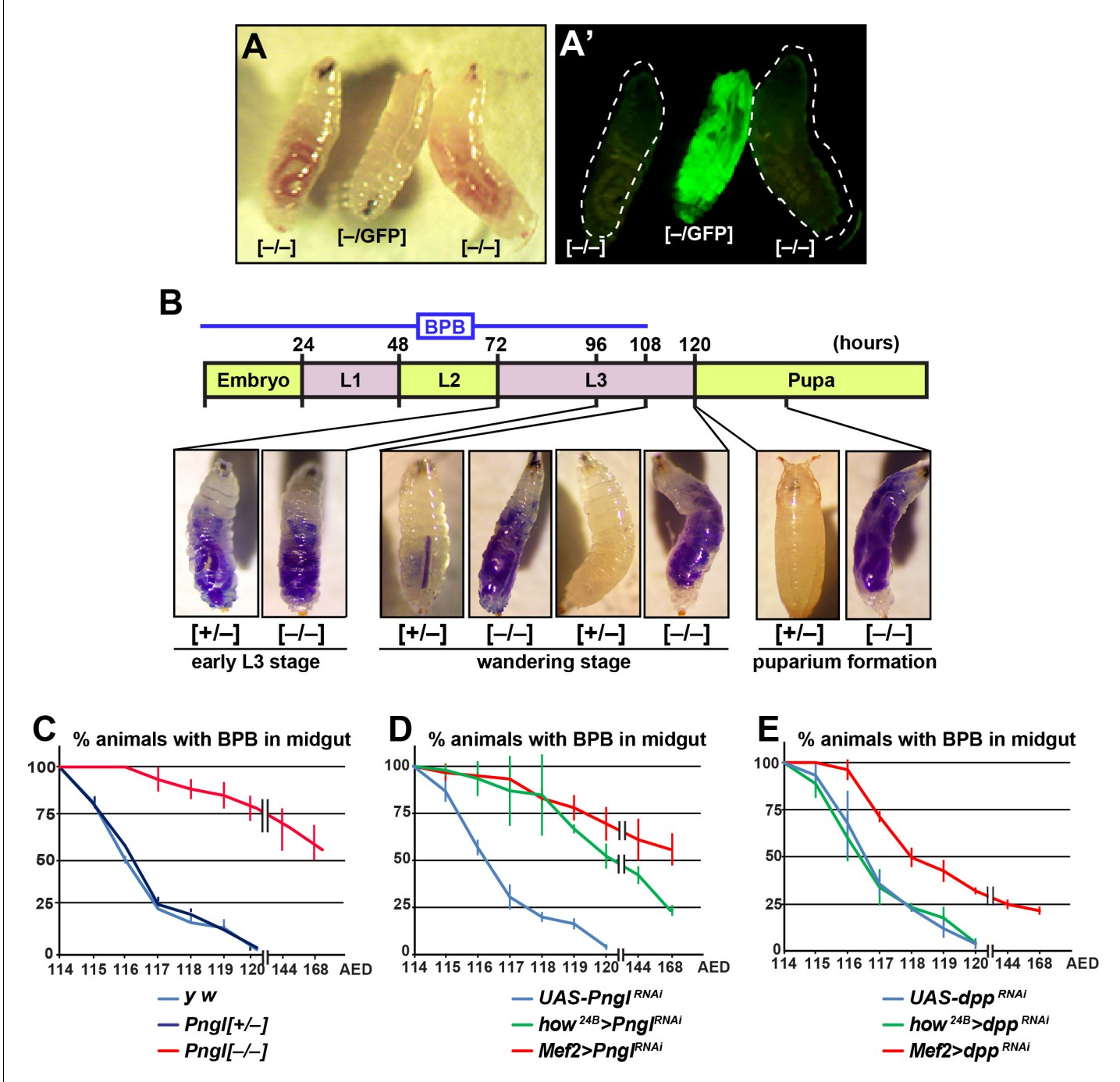

**Figure 4.** Loss of *Pngl* in the mesoderm causes food accumulation in larval midgut. (**A–A'**) Wandering larvae of *Pngl*[ex14/+] with a GFP[+] balancer chromosome for selection [–/GFP] and *Pngl*[ex14/ex14] GFP[–] larvae [–/–] showing food accumulation. (**B**) Gut clearance assay in larval stages using Bromophenol blue (BPB) as a marker. The top part shows the timeline of BPB feeding during development. At the bottom, representative images of *Pngl*[ex14/+] [+/–] and *Pngl*[ex14/ex14] larvae [–/–] at different times and stages are shown. (**C–E**) Quantification of midgut clearance assays in third instar larvae of the indicated genotypes.

DOI: https://doi.org/10.7554/eLife.27612.009

(BPB), they show a dark blue gut approximately 24–48 hours before puparium formation (early L3 stage). At late L3 stage, they stop eating and enter the wandering stage, during which they empty their gut and reach the stationary stage (*Denton et al., 2008*). *Pngl*[+/–] (control) and *Pngl*[–/–] larvae both exhibit a blue gut during early L3 stage, indicative of eating food containing BPB (*Figure 4B*).

Once taken off the BPB food during the wandering stage, control larvae gradually lose the blue color and pupariate, but the *Pngl*$^{-/-}$ larvae fail to empty their guts and exhibit a delay in pupariation (*Figure 4B*). Control *y w* and *Pngl*$^{+/-}$ larvae typically emptied their gut in 6 hours and reached the pre-pupal stage (*Figure 4C*). In contrast, most *Pngl*$^{-/-}$ larvae showed a blue gut throughout the wandering stage (*Figure 4C*). Indeed, even 24–48 hours later, more than 50% of the *Pngl*$^{+/-}$ animals were still in the larval stage and retained the blue food in their abdomens. These data indicate that loss of *Pngl* causes an impairment of gut clearance at late L3 stage.

To examine whether impaired BMP signaling can also explain the food accumulation phenotype observed upon loss of *Pngl*, we compared the *dpp* and *Pngl* KD phenotypes by the same mesodermal drivers. BPB feeding assays showed that *Mef2 > dpp*$^{RNAi}$ larvae exhibit a food accumulation phenotype milder than that observed in *Pngl* mutant and *Mef2 > Pngl*$^{RNAi}$ larvae, and that *how*$^{24B}$ > *dpp*$^{RNAi}$ larvae do not exhibit any food accumulation phenotype, in contrast to *how*$^{24B}$ > *Pngl*$^{RNAi}$ animals (*Figure 4D*, compare to *Figure 4E*). These data indicate that the food accumulation phenotype observed in *Pngl* mutants is at least in part independent of impaired BMP signaling.

## Pngl is required for proper Dpp propagation and autoactivation in the embryonic VM

Loss of *Pngl* affects BMP signaling in the midgut but not in the ectodermal regions of the *Drosophila* embryos. To shed light on the mechanism by which *Pngl* regulates BMP signaling in a tissue-specific manner, we stained *Pngl* mutant and control embryos with a polyclonal anti-Dpp antibody raised against its prodomain (*Akiyama and Gibson, 2015*). In WT and *Pngl*$^{+/-}$ embryos at stage 13, the Dpp protein is expressed in ectodermal bands, several regions in the anterior part of the embryo and two rather narrow groups of cells corresponding to PS3 and PS7, where it shows a punctate pattern superimposed on a diffuse signal (*Figure 5A, B, G and H*). Overall, the staining is somewhat weaker in *Pngl*$^{-/-}$ embryos, but staining in PS3 and PS7 is observed, with a notable decrease in puncta (*Figure 5C and I*). By stage 14, a broad and strong Dpp expression domain can be observed in PS3 and PS7 in control embryos (*Figure 5D, E, J and K*). In contrast, Dpp expression in the VM remains weak and narrow in PS3 and PS7 of mutant embryos (*Figure 5F and L*). Notably, the pattern of Dpp expression in the ectoderm is similar in control and *Pngl*$^{-/-}$ embryos at both stages, even though the staining seems to be slightly weaker in mutant embryos (*Figure 5A–F*). These results suggest that a failure to expand the Dpp expression domain in the VM leads to impaired mesoderm-to-endoderm BMP signaling in *Pngl* mutants.

Given the weaker Dpp staining in stage 13 *Pngl* mutant embryos, we asked whether the failure to expand the Dpp expression domain and loss of BMP signaling in the embryonic midgut of *Pngl* mutants is simply due to reduction of *dpp* expression in the mesoderm. To address this point, we used the *dpp*$^{S2}$ allele, which harbors a chromosomal breakpoint in the regulatory elements that control *dpp* expression in digestive tract (*Masucci and Hoffmann, 1993*). Embryos heterozygous for this allele exhibit a decrease in Dpp expression at stage 13 in PS7, even compared to *Pngl*$^{-/-}$ embryos (*Figure 5M*, compare to *Figure 5G, H and I*). Nevertheless, these embryos show Dpp propagation at stage 14 and proper Dpp signaling from mesoderm to endoderm, as evidenced by analyzing the expression of Dpp targets in the embryonic midgut and gastric caeca and acid zone in larvae (*Figure 5N* and *Figure 5—figure supplement 1*). These data suggest that decreased Dpp expression by itself cannot explain the impaired Dpp propagation and loss of mesoderm-to-endoderm BMP signaling in *Pngl* mutants.

To understand how Dpp spreads in PS7 during embryonic midgut development, we expressed *UAS-GFP* in embryonic mesoderm by *Mef2-GAL4* driver and stained embryos for pMad at stage 13 and 14. At stage 13 the majority of pMad staining is localized to the VM and only a few cells are stained in the endoderm (*Figure 5O and O'*). Later at stage 14, BMP signaling spreads through the endoderm and expands in PS7, as evidenced by the presence of many pMad-positive endodermal cells (*Figure 5P and P'*; GFP$^-$, round nuclei). These observations are in agreement with previous reports suggesting the existence of an autoregulatory Dpp para-autocrine loop in PS7, which acts through *Ultrabithorax* (*Ubx*) expression in VM and results in maintenance and accumulation of Dpp expression in VM cells (*Figure 5Q*) (*Bienz, 1997*; *Hursh et al., 1993*; *Staehling-Hampton and Hoffmann, 1994*). To test whether Pngl affects Dpp autoactivation in VM, we used a *dpp* enhancer trap line and stained embryos for pMad and βGAL. In *Pngl*$^{+/-}$ embryos at stage 13, we found a row of

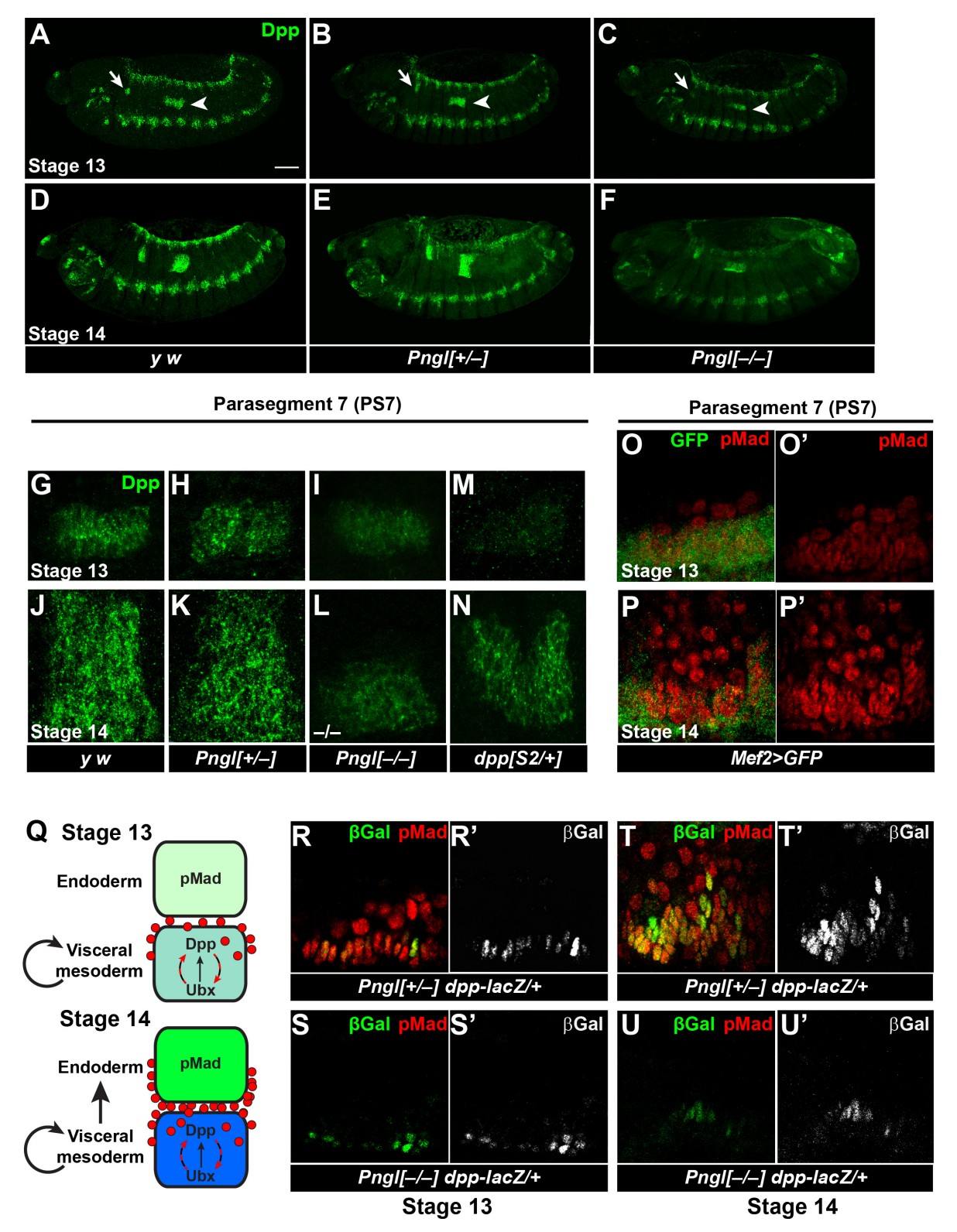

**Figure 5.** *Pngl* is required for proper Dpp propagation and autoactivation in the embryonic VM. (A–F) Projection views of Dpp staining for stage 13 (A–C) and stage 14 (D–F) embryos of the indicated genotypes are shown. Arrows and arrowheads mark PS3 and PS7, respectively. Scale bar in A is 100 µm. (G–N) Dpp staining in PS7. Limited projection views for stages 13 (G–I and M) and 14 (J–L and N) embryos of the indicated genotypes are shown. (O–P') pMad staining of PS7 at stages 13 (O–O') and 14 (P–P'). Green marks mesoderm (*Mef2 >GFP*). (Q) Schematic drawing of Dpp autoactivation in PS7

*Figure 5 continued on next page*

*Figure 5 continued*

in stages 13 and 14. (**R–U'**) Limited projection views of PS7 for the indicated genotypes at embryonic stages 13 and 14. *dpp-lacZ* is marked in green (gray in R'-U'), red marks pMad.

DOI: https://doi.org/10.7554/eLife.27612.010

The following figure supplements are available for figure 5:

**Figure supplement 1.** Removing one copy of *dpp* in the visceral mesoderm does not affect BMP signaling in PS7.

DOI: https://doi.org/10.7554/eLife.27612.011

**Figure supplement 2.** Dpp signaling in embryonic dorsal ectoderm is not impaired in *Pngl* mutants, but the Dpp-positive puncta are severely decreased.

DOI: https://doi.org/10.7554/eLife.27612.012

cells co-expressing pMad and βGAL (*Figure 5R–R'*), suggesting para-autocrine activation. In PS7 of *Pngl$^{-/-}$* embryos at stage 13, *dpp-lacZ* expression can be detected, but pMad staining is absent (*Figure 5S–S'*). These results indicate that in *Pngl$^{-/-}$* PS7, the initial, Ubx-dependent *dpp* expression (*Sun et al., 1995*) occurs, but the autoactivation loop is impaired. At stage 14, the Dpp expression domain expands in *Pngl$^{+/-}$* embryos and *dpp-lacZ* expressing cells can be seen in multiple layers. Most if not all *dpp-lacZ* expressing cells still co-express pMad, likely due to autoactivation, while endodermal cells, as receiving cells, only express pMad (*Figure 5T–T'*). In stage 14 *Pngl$^{-/-}$* embryos, *dpp-lacZ*-expressing cells remain confined to their initial narrow domain in PS7, and pMad staining is still not detectable (*Figure 5U–U'*). This is in contrast to the dorsal ectodermal region, which showed a comparable expression pattern and intensity in the dorsal ectodermal bands of *Pngl$^{+/-}$* and *Pngl$^{-/-}$* embryos for both *dpp-lacZ* and pMad (*Figure 5—figure supplement 2*). Altogether, these data indicate that *Pngl* is not required for the initial expression of *dpp* through *Ubx* in *Drosophila* embryos but is required for the autoregulatory role of Dpp in the VM.

## Loss of BMP signaling in *Pngl$^{-/-}$* embryonic endoderm is caused by impaired BMP autoactivation in VM

We next asked whether bypassing the BMP autoactivation loop by mesodermal overexpression of a GFP-tagged version of Dpp (*Teleman and Cohen, 2000*) can rescue *Pngl* phenotypes in embryos. In control animals, *dpp-GFP* overexpression by *Mef2-GAL4* resulted in broad pMad staining in embryonic midgut and ectopic pMad expression throughout the ectoderm (*Figure 6A and D*). Dpp-GFP protein can be seen by GFP staining and shows a continuous pattern going through PS3 to PS7 (*Figure 6D'*). In *Pngl$^{-/-}$* embryos, mesodermal expression of Dpp-GFP restored pMad expression in PS3 and PS7, indicating a rescue of BMP signaling from VM to endoderm (*Figure 6B and E*, compare to *Figure 6C and F*). Notably, in *Pngl$^{-/-}$; Mef2 > dpp* GFP embryos, pMad staining was limited to PS3 and PS7, and ectopic pMad expression in the ectoderm was almost completely suppressed (*Figure 6B and E*), indicating that loss of *Pngl* dramatically decreases the ability of mesodermally-expressed Dpp-GFP to induce ectopic BMP signaling.

If BMP autoactivation is impaired in *Pngl$^{-/-}$* VM, bypassing the normal autoactivation process by expressing a constitutively active form of the BMP receptor Tkv (*tkv$^{CA}$*) in the mesoderm should restore BMP signaling in the PS3 and PS7 endoderm. *Mef2 > tkv$^{CA}$* embryos exhibited proper pMad staining in PS3 and PS7 in a *Pngl$^{+/+}$* background, although some pMad staining outside of PS7 could be seen in the VM marked by Fas3 (*Figure 6G and G'*, arrows). When *tkv$^{CA}$* is overexpressed in *Pngl$^{-/-}$* embryonic mesoderm, pMad was restored in PS3 and PS7 and still some extra pMad positive cells were detected (*Figure 6H and H'*, arrows). Moreover, ~30% of *Pngl$^{-/-}$; Mef2 > tkv$^{CA}$* animals reach adulthood (*Figure 6I*). These observations further indicate that impaired BMP signaling is partially responsible for the lethality of *Pngl$^{-/-}$* animals. This is likely due to gastric caeca shortening (*Masucci and Hoffmann, 1993*), although we cannot exclude that *tkvCA* overexpression affects unknown BMP-related defects in other parts of mesoderm in these mutants. We note that the rescued *Pngl$^{-/-}$; Mef2 > tkv$^{CA}$* adults did not show the acid zone loss and midgut shortening phenotypes observed in *Pngl$^{-/-}$* adult escapers (*Figure 6—figure supplement 1*). This indicates that the BMP-related midgut phenotypes of *Pngl* mutants persist to adulthood. Altogether, these results support the notion that BMP signaling in VM requires an autoregulatory step to reach the required threshold for signaling to endoderm and that this is the step during midgut development that is

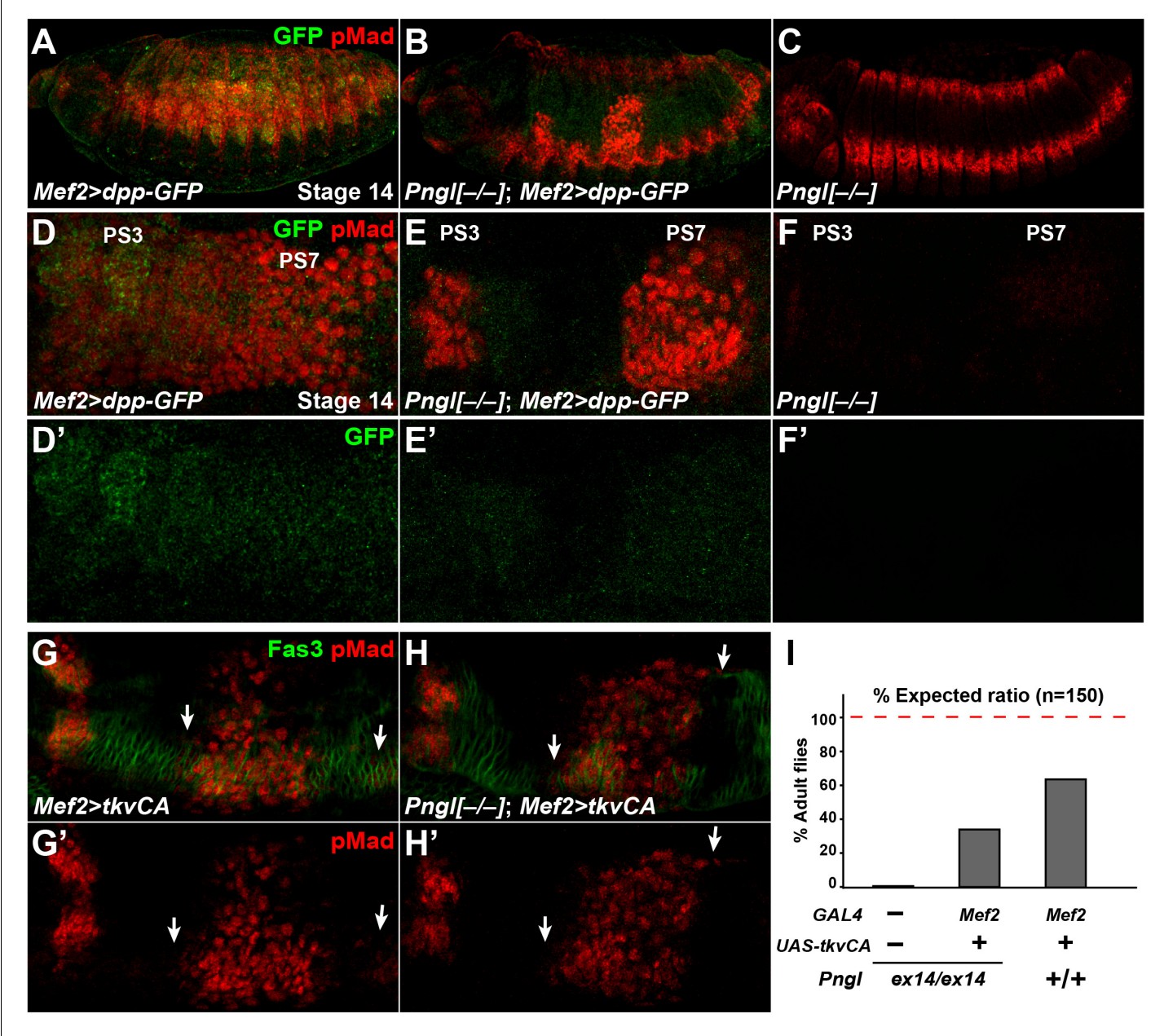

**Figure 6.** Dpp-GFP and Tkv[CA] overexpression in mesoderm rescues Dpp signaling in *Pngl*[−/−] embryo. (**A–C**) Projection views of stage 14 embryos of the indicated genotypes stained with anti-pMad (red) and anti-GFP (green) antibodies. (**D–F'**) Close-ups of embryonic PS3 and PS7 of indicated genotypes. Notably, in the *Pngl*[−/−] embryo, *Mef2 >dpp* GFP overexpression rescues pMad staining in PS3 and PS7 (E, compared to F). (**G–H'**) Close-ups of embryonic PS3 and PS7 of indicated genotypes. pMad is marked in red and Fas3 in green. Arrows mark pMad staining outside of PS7. (**I**) Eclosion tests of the indicated genotypes. The red dashed line marks the expected Mendelian ratio.

DOI: https://doi.org/10.7554/eLife.27612.013

The following figure supplement is available for figure 6:

**Figure supplement 1.** The acid zone impairment in adult midgut of Pngl mutant flies can be rescued by overexpressing *tvkCA* in the mesoderm.

DOI: https://doi.org/10.7554/eLife.27612.014

impaired in *Pngl* mutants. The data also indicate that this developmental defect contributes to the semi-lethality of *Pngl* mutants.

## BMP autoactivation in VM is mediated by Dpp and Tkv homodimers

BMPs can function as homodimers or heterodimers to signal through homo- and/or heterodimeric type I receptors (*Hogan, 1996*; *O'Connor et al., 2006*). To date, three BMP ligands have been identified in *Drosophila*: *dpp*, *glass bottom boat* (*gbb*) and *screw* (*scw*) (*Arora et al., 1994*; *Doctor et al., 1992*; *Padgett et al., 1987*; *Wharton et al., 1999*). *scw* is required in early embryogenesis, where it acts in combination with *dpp* to specify dorsal cells fates (*Arora et al., 1994*). *gbb* and *dpp* mutants have both distinct and overlapping phenotypes during and after embryonic development, suggesting that *gbb* and *dpp* are required together for proper signaling in some developmental contexts (*Goold and Davis, 2007*; *Haerry et al., 1998*; *Hong et al., 2016*; *Khalsa et al., 1998*; *Wharton et al., 1999*). To assess whether *gbb* is involved in BMP signaling and Dpp expression in embryonic midgut and the generation of $Pngl^{-/-}$ phenotypes, we performed Dpp and pMad staining in *gbb* mutant embryos at stage 14, focusing on PS3 and PS7. Unlike control embryos (*Figure 7A* and *Figure 7—figure supplement 1*), $gbb^{D4/D20}$ and $gbb^{D4/D4}$ embryos showed reduced pMad staining in PS3 (*Figure 7C* and *Figure 7—figure supplement 1*), in agreement with previous work indicating that gastric caeca are somewhat short in *gbb* mutant larvae (*Wharton et al., 1999*). However, in $gbb^{-/-}$ embryos pMad staining in PS7 is increased and shifts anteriorly towards PS3 (*Figure 7C* and *Figure 7—figure supplement 1*). In $gbb^{D4/D20}$ embryos, the Dpp expression domain in PS3 is similar in size to that in control embryos (*Figure 7—figure supplement 2*). However, the Dpp expression domain in PS7 is broader and stronger than controls in these embryos and extends anteriorly to PS6, and additional Dpp-positive puncta can be detected outside of the acid zone region towards PS3 (*Figure 7—figure supplement 2*). At stage 16, $gbb^{-/-}$ embryos show an abnormal expansion of the second midgut compartment and a diffuse Labial staining both anteriorly and posteriorly to the second constriction (*Figure 7D and D'* and *Figure 7—figure supplement 1*) compared to controls (*Figure 7B and B'* and *Figure 7—figure supplement 1*). Therefore, *gbb* phenotypes in the midgut are quite different from *dpp* and *Pngl* loss-of-function phenotypes. The strong and broader than normal pMad expression domain in $gbb^{-/-}$ PS7 indicates that BMP signaling in this region is mediated by Dpp homodimers alone and Gbb likely limits the area in the VM wherein BMP autoactivation and signaling can occur.

In *Drosophila*, two type I receptors Tkv and Saxophone (Sax) and one type II receptor (Punt) are involved in BMP signaling (*Letsou et al., 1995*; *Nellen et al., 1994*; *Nellen et al., 1996*; *Ruberte et al., 1995*). Given the differential roles of the Dpp versus Gbb ligands in BMP autoactivation in VM, we asked whether the contribution of Tkv and Sax to this process is also different from each other. As shown in *Figure 6E*, mesodermal knock-down of *tkv* results in a severe reduction of pMad staining in PS7 and a partial loss of pMad in PS3. Similar to $Pngl^{-/-}$ embryos, the Dpp expression domain in PS7 fails to expand in $Mef2 > tkv^{RNAi}$ embryos, although *tkv* KD does not seem to affect the intensity of Dpp staining at stage 13 or the presence of Dpp-positive puncta (*Figure 7—figure supplement 2*). In $Mef2 > sax^{RNAi}$ embryos, pMad staining in PS3 seems to be slightly decreased compared to controls, but it looks broader in PS7 and is extended anteriorly (*Figure 7G*). Furthermore, $Mef2 > tkv^{RNAi}$ embryos show a lack of second constriction and loss of Labial in the PS7 region (*Figure 7F and F'*). In contrast, mesodermal knock-down of *sax* did not show evident midgut constriction defects, but aberrant Labial staining was detectable both anteriorly and posteriorly to the second constriction, similar to $gbb^{-/-}$ embryos (*Figure 7H and H'*). These results strongly suggest that BMP autoactivation in PS7 mesoderm is mediated by Dpp homodimers, as ligand, and Tkv homodimers, as receptor.

Similar to its vertebrate homologs, Dpp is synthetized as an inactive proprotein (*Figure 7I*). Following dimerization in the endoplasmic reticulum (ER), Dpp proprotein dimers traffic through the secretory pathway, where they undergo cleavages to release the active dimer (*Christian, 2012*; *Künnapuu et al., 2009*; *Sopory et al., 2010*). Since our results indicate a role for *Pngl* in Dpp autoactivation mediated by Dpp homodimers (as opposed to Dpp-Gbb heterodimers) in PS7 VM, we examined the effects of loss of Pngl on Dpp dimerization and processing by Western blotting on larval extracts using the above-mentioned anti-Dpp antibody (*Akiyama and Gibson, 2015*). When ran on a reducing gel, control larvae (*y w* and *UAS-Pngl^{RNAi}* without a GAL4 driver) exhibited two prominent bands: one corresponding in size to the full-length proprotein monomer and the other to the proprotein dimer, suggesting that Dpp dimers are partially resistant to the amount of reducing agents used in our assays (*Figure 7J*). A third band slightly bigger and much fainter than the

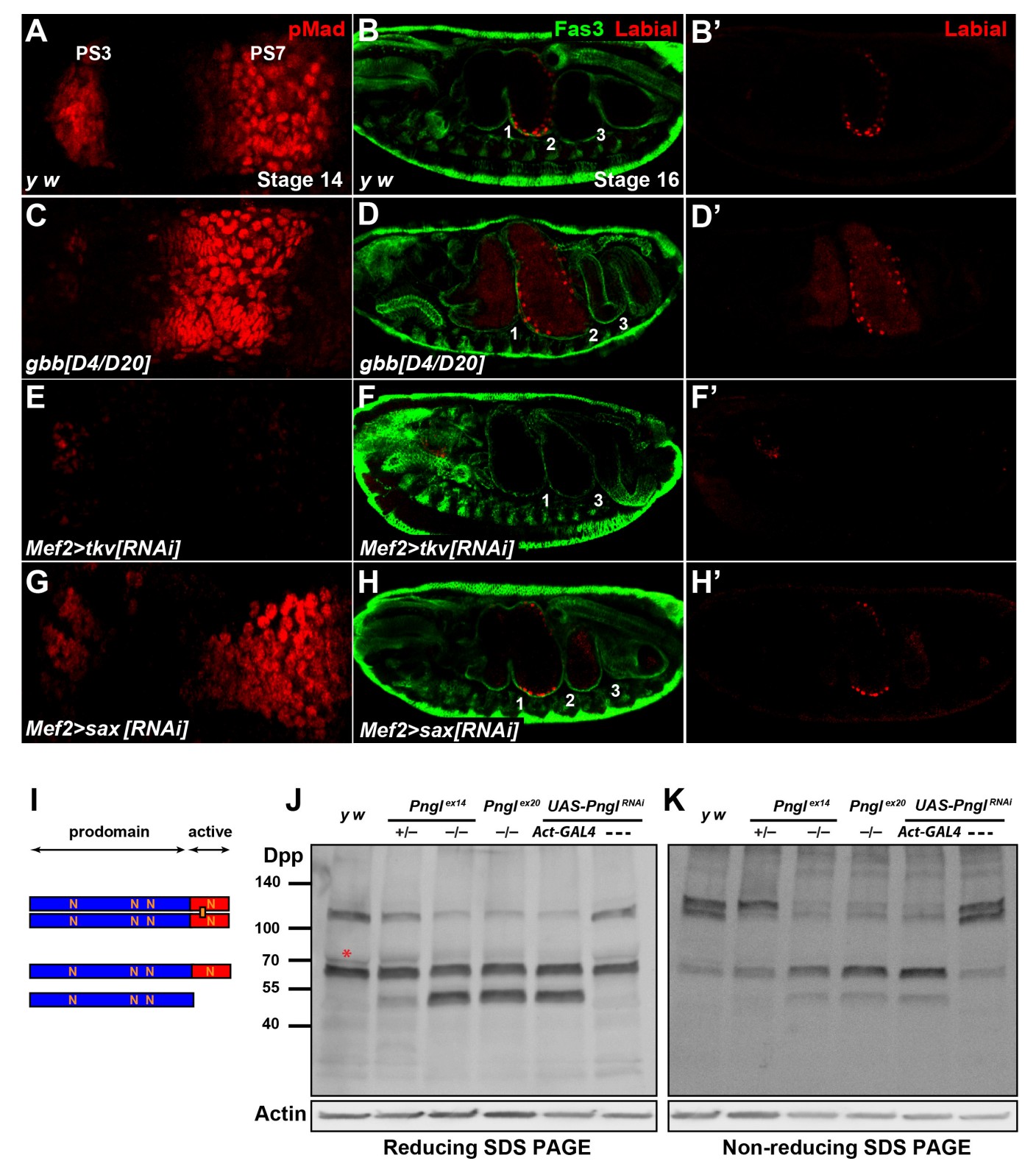

**Figure 7.** Pngl regulates Dpp homodimer level and signaling via Tkv receptor homodimers. (**A, C, E, G**) Close-ups of embryonic PS3 and PS7 of indicated genotypes. pMad is marked in red. (**B–B', D–D', F–F' and H–H'**) Fas3 (VM marker) and Labial staining of stage 16 embryos of indicated genotypes. Numbers indicate embryonic midgut constrictions. (**I**) Schematic representation of the Dpp protein dimer and its processing. (**J, K**)

*Figure 7 continued on next page*

*Figure 7 continued*

Reducing (**J**) and non-reducing (**K**) SDS gels were used to run larval extracts from the indicated genotypes and were probed with a polyclonal antibody against the Dpp prodomain.

DOI: https://doi.org/10.7554/eLife.27612.015

The following figure supplements are available for figure 7:

**Figure supplement 1.** Loss of *gbb* affects BMP signaling in embryonic midgut differently from loss of *dpp* or *Pngl*.

DOI: https://doi.org/10.7554/eLife.27612.016

**Figure supplement 2.** Altered Dpp expression in *gbb*[−/−] and *tkv* mesodermal KD embryos.

DOI: https://doi.org/10.7554/eLife.27612.017

**Figure supplement 3.** Dpp dimer levels are not exclusively altered in *Pngl* larval midgut.

DOI: https://doi.org/10.7554/eLife.27612.018

proprotein monomer was also observed in control larvae (*Figure 6J*, red asterisk). Larvae homozygous for *Pngl* alleles and those ubiquitously expressing *Pngl* dsRNA showed an accumulation of a band corresponding in size to the prodomain (cleavage product) and a significant decrease in the intensity of the band corresponding to Dpp dimers, without an apparent decrease in the intensity of the proprotein monomer band (*Figure 7J*). Of note, although larvae heterozygous for the *Pngl*[ex14] allele did not show any midgut defects (*Figure 2—figure supplement 1*), they displayed an intermediate pattern in the Western blot, suggesting that Pngl plays a dosage-sensitive role in Dpp dimerization and/or processing (*Figure 7J*). To better assess the level of Dpp dimers in a *Pngl*-deficient background, we performed Western blot analysis on larval extracts ran on a non-reducing SDS gel electrophoresis (*Figure 7K*). Control larvae showed two rather strong bands corresponding in size to Dpp dimers and a weaker band corresponding to a proprotein monomer. This indicates that in a non-denaturing state, Dpp molecules are primarily found in association with other molecules and form Dpp-Dpp homodimers. We found a significant decrease in Dpp dimers in *Pngl* mutant and knock-down larvae and a corresponding increase in the level of full-length monomer and the same cleavage product observed in regular Western blots. Again, *Pngl*[ex14/+] larvae showed an intermediate band pattern, with a significant decrease in the intensity of one of the two 'dimer' bands and an apparent increase in the intensity of the full-length monomer (*Figure 7K*). Dpp dimers are not exclusively reduced in midguts of *Pngl* mutants but are also decreased in other parts of the larval body (*Figure 7—figure supplement 3*). In summary, these data provide strong evidence that Pngl is required for the formation and/or the stability of Dpp dimers.

## Discussion

The broad phenotypes of children affected with NGLY1 deficiency (*Enns et al., 2014*; *Need et al., 2012*) and the semi-lethality of *Pngl*[−/−] flies (*Funakoshi et al., 2010*) indicate that NGLY1 plays important roles during animal development. However, the *N*-glycanase function has not been linked to any developmental signaling pathway. Here we report that fly *Pngl* regulates BMP signaling during embryonic midgut development without affecting BMP signaling in ectodermal and head regions of the embryo. Our data indicate that *Pngl* is not required in the midgut endoderm to receive the BMP signal, but rather is required in the VM to send the BMP signal. It has previously been shown that BMP signaling uses a paracrine/autocrine loop in the VM to sustain and increase the expression of Dpp in PS3 and PS7 of embryonic VM. This loop is proposed to ensure that the level of BMP ligands in the VM is high enough to induce signaling in the endoderm and to specify gastric caeca, the second midgut constriction and the acid zone (*Bienz, 1997*; *Hursh et al., 1993*; *Staehling-Hampton and Hoffmann, 1994*). Several lines of evidence indicate that the BMP autoregulation mediated by the para-autocrine loop in the VM is the step which is impaired in *Pngl*-deficient embryos. First, Pngl is not required for the initial, Ubx-dependent expression of *dpp*. In fact, even a 50% decrease in the expression of *dpp* in the visceral mesoderm of *dpp*[s2/+] animals does not impair BMP autoactivation and midgut development. Second, despite expressing Dpp at early stages, BMP signaling is not activated in *Pngl*[−/−] VM, as evidenced by the lack of pMad staining. Third, overexpression of Dpp-GFP in the mesoderm is able to induce BMP signaling in the endoderm in *Pngl*[−/−] embryos. Lastly, bypassing the para-autocrine loop by transgenic expression of a constitutively

active BMP receptor in the mesoderm results in restoration of BMP signaling in PS3 and PS7 regions of the endoderm and in partial rescue of lethality in $Pngl^{-/-}$ embryos.

In the BMP para-autocrine loop, VM cells both secrete the BMP ligand and respond to it. Therefore, theoretically, Pngl might play a critical role in sending the BMP signal, receiving the BMP signal, or both. Although our data do not allow us to exclude any of these possibilities, based on the following observations, we favor a scenario in which Pngl is required in VM cells to send the Dpp signal not to receive it: (1) Pngl is not required to receive the BMP signal in the endoderm; (2) Loss of Pngl and *Pngl* KD result in a dramatic decrease in the level of Dpp homodimers and the Dpp-positive puncta; (3) Expression of a constitutively active form of Tkv in the mesoderm is able to restore midgut pMad staining in embryos and the copper cell region in the adult midgut, and partially rescue the lethality of $Pngl^{-/-}$ animals; (4) Loss of *Pngl* almost fully suppresses the aberrant BMP signaling caused by mesodermal overexpression of Dpp-GFP.

Whole larval protein extracts from *Pngl*-deficient animals show an increase in the level of the monomeric forms of Dpp (full-length and a cleavage product) and a simultaneous decrease in the bands corresponding in size to Dpp dimers. Moreover, $Pngl^{-/-}$ embryos show a decrease in Dpp-positive puncta both in the mesoderm, where signaling is impaired, and in the ectoderm, where signaling is not impaired. Together, these observations indicate that the effect of loss of Pngl on the Dpp protein itself is not limited to the mesoderm. Indeed, protein extracts from *Pngl*-deficient midgut and carcass (without midgut) both show a decrease in Dpp dimer levels. This suggests that either Pngl regulates BMP signaling by affecting Dpp dimer levels in other larval tissues not identified yet, or that Dpp dimers are only important in the midgut and although they are decreased elsewhere, Dpp-Gbb heterodimers compensate for the lack of Dpp dimers in most other tissues. Regardless, we propose that loss of BMP signaling in *Pngl* mutant midguts results from a requirement for Dpp homodimers in the para-autocrine autoregulatory loop present in the visceral mesoderm.

BMP ligands can signal both as homodimers and as heterodimers (*Bragdon et al., 2011*; *O'Connor et al., 2006*). *In vitro* and *in vivo* studies have shown that in general, BMP heterodimers have stronger bioactivity than their homodimers counterparts (*Aono et al., 1995*; *Butler and Dodd, 2003*; *Israel et al., 1996*; *Little and Mullins, 2009*; *Morimoto et al., 2015*; *Valera et al., 2010*). In some cases, the homodimers induce weak to moderate signaling, and in other cases they either do not elicit signaling or even play an antagonistic role (*Bangi and Wharton, 2006b*; *O'Connor et al., 2006*). Stronger activity of BMP heterodimers can at least in part be explained by differential affinities of individual BMP ligands for different BMP receptors, combined with stronger signal transduction by heterodimeric type I receptors compared to homodimers of each type I receptor. For example, in *Drosophila*, Dpp has a higher affinity for Tkv, whereas the other two ligands–Gbb and Scw–have a higher affinity for Sax (*Bangi and Wharton, 2006b*; *O'Connor et al., 2006*; *Shimmi et al., 2005*). A similar receptor-ligand binding preference has been observed among the vertebrate orthologs (*Aoki et al., 2001*; *Little and Mullins, 2009*). In the embryonic dorsal midline and the wing imaginal disc, Dpp/Scw and Dpp/Gbb heterodimers induce high levels of signaling, respectively, through Tkv/Sax heterodimers (*Bangi and Wharton, 2006b*; *O'Connor et al., 2006*). Comparison of the *gbb* mutant phenotypes in the midgut with those caused by *Pngl* loss and by *dpp* KD indicates that Dpp homodimers are the only productive form of ligand in PS7. Moreover, mesodermal KD of *tkv* severely decreases BMP signaling in PS7, but mesodermal KD of *sax* not only does not decrease pMad staining in PS7, but also results in an expansion of pMad expression domain in the PS7 region, similar to *gbb* mutant embryos. Together, these observations strongly support the notion that the BMP autoregulatory loop in the VM, which is essential for the activation of BMP signaling in the endoderm, relies solely on Dpp and Tkv homodimers, and therefore is impaired in *Pngl* mutants due to the severe decrease in the level of Dpp homodimers in these animals.

Vertebrate and invertebrate BMP proteins and other members of the TGFβ superfamily each harbor several *N*-linked glycosylation sites, which have been shown to be glycosylated in many cases (*Groppe et al., 1998*; *Miyazono and Heldin, 1989*; *Tauscher et al., 2016*). Various functional roles have been ascribed to *N*-glycans on these ligands, including enhancing receptor binding of BMP6 (*Saremba et al., 2008*), keeping the TGFβ1 ligand in a latent state (*Miyazono and Heldin, 1989*), and promoting inhibin (α/β) heterodimer formation at the expense of activin (β/β) homodimer formation (*Antenos et al., 2007*). Accordingly, given the significant increase in Dpp monomeric forms and

the simultaneous decrease in Dpp dimers upon loss of *Pngl*, it is possible that Pngl removes one or more *N*-glycans from Dpp and thereby promotes the formation or the stability of Dpp homodimers. Whether the regulation of Dpp by Pngl is direct or mediated via other proteins will remain to be explored.

In agreement with a previous report (*Masucci and Hoffmann, 1993*), our data suggest that the lethality of *Pngl* mutants cannot be fully explained by shortening of the gastric caeca and impairment of BMP signaling in midgut development. *Pngl* KD with mesodermal drivers leads to a higher degree of lethality compared to *dpp* KD with the same drivers. Moreover, $how^{24B} > Pngl^{RNAi}$ animals show ~70% lethality, even though they do not have gastric caeca defects. Finally, restoring BMP signaling in the midgut by expressing *tkvCA* only recues the lethality in ~30% of $Pngl^{-/-}$ animals. Phenotypic analysis of *Pngl* mutants combined with rescue and KD experiments suggest that a failure to properly empty the gut before puparium formation contributes to lethality in these animals. The molecular mechanisms for the food accumulation phenotype and other potential $Pngl^{-/-}$ phenotypes contributing to lethality are still under investigation.

In summary, our work indicates that the fly Pngl is an evolutionarily conserved *N*-glycanase enzyme necessary to sustain BMP autoactivation in the VM mediated by para-autocrine activity of Dpp homodimers through Tkv homodimers. Although we cannot exclude that Pngl plays important roles in other cell types as well, our data indicate that Pngl is primarily required in the mesoderm during midgut development and its loss results in Dpp-dependent and Dpp-independent midgut defects. Given the reports on potential para-autocrine functions of mammalian Dpp homologs (*Grimsrud et al., 1999*; *Rege et al., 2015*; *Shukunami et al., 2000*; *Tokola et al., 2015*) and prominent human pathologies associated with dysregulated BMP signaling in ophthalmic, gastrointestinal and musculoskeletal systems (*Wang et al., 2014*), tissue-specific alterations in BMP signaling might contribute to some of the NGLY1 deficiency phenotypes including retinal abnormalities, delayed bone age and osteopenia, small feet and hands, and chronic constipation (*Enns et al., 2014*; *Lam et al., 2017*). Finally, understanding the mechanisms underlying the food accumulation phenotype in $Pngl^{-/-}$ larvae might shed light on the pathophysiology of chronic constipation in NGLY1 deficiency patients.

## Materials and methods

### *Drosophila* strains and genetics

Animals were grown on standard food containing cornmeal, molasses and yeast at room temperature, except for RNAi crosses, which were cultured at 30°C. The following strains were used in this study: *y w, w; L/CyO, kr-GAL4 UAS-GFP* (CyO-GFP), *TM3, Sb¹/TM6, Tb¹, Act-GAL4, Mef2-GAL4, how²⁴ᴮ-GAL4* (*Brand and Perrimon, 1993*; *Staehling-Hampton et al., 1994*), $dpp^{10638}$ (*dpp-lacZ*) (*Zecca et al., 1995*), $dpp^{s2}$ (*Masucci and Hoffmann, 1993*), *UAS-dpp-GFP* (*Teleman and Cohen, 2000*), $UAS-dpp^{RNAi}$ (*Liu et al., 2010*), $UAS-tkv^{RNAi}$, $gbb^{D4}$ and $gbb^{D20}$ (*Chen et al., 1998*), $UAS-tkv^{CA}$ (*Adachi-Yamada et al., 1999*), *Df(2R)ED1484, UAS-CD8::GFP* (Bloomington *Drosophila* Stock Center), $Pngl^{ex14}$, $Pngl^{ex18}$, $Pngl^{ex20}$, *UAS-Pngl* and *UAS-Pngl-C303A* (*Funakoshi et al., 2010*), *NP3207-GAL4* and *NP3270-GAL4* (*Tanaka et al., 2007*) (Kyoto *Drosophila* Stock Center), $UAS-Pngl^{RNAi}$ KK101641, $UAS-sax^{RNAi}$ (Vienna *Drosophila* Resource Center), *UAS-attB-NGLY1-WT-VK31* and *UAS-attB-NGLY1-ΔR402-VK31* (this study).

To identify homozygous animals in sibling crosses, *Pngl* and *gbb* mutants were balanced over a *CyO, kr-GAL4 UAS-GFP* chromosome. To examine *dpp* transcription in a $Pngl^{-/-}$ background, a *dpp-lacZ* $Pngl^{ex14}$/CyO-GFP recombinant strain was generated and crossed to $Pngl^{ex14}$/CyO GFP animals. To overexpress NGLY1 in $Pngl^{-/-}$ animals, $Pngl^-$ (ex14 or ex18)/CyO; GAL4 (Actin, Mef2 or $how^{24B}$)/TM6, $Tb^1$ animals were crossed to $Pngl^-$/CyO; UAS-attB-NGLY1-VK31 (WT or ΔR402)/TM6, $Tb^1$ animals. To overexpress Tkv$^{CA}$ or Dpp-GFP in a $Pngl^{-/-}$ background, $Pngl^{ex14}$/CyO-GFP; Mef2-GAL4 animals were crossed to $Pngl^{ex14}$/CyO-GFP; UAS-tkv$^{CA}$ (or UAS-dpp-GFP) and absence of the CyO-GFP was used to select the intended genotype. To overexpress Pngl and Pngl-C303A in $Pngl^{-/-}$ animals, $Pngl^{ex14}$/CyO-GFP; Mef2-GAL4 animals were crossed to $Pngl^{ex14}$/CyO-GFP; UAS-Pngl and $Pngl^{ex14}$/CyO-GFP; UAS-Pngl-C303A animals and selected similar to the above crosses.

## Survival, longevity, and fertility assays

For survival (eclosion) tests, the expected ratio of offspring was calculated based on Mendelian inheritance for each genotypic class and the observed/expected ratio is reported as a percentage. Fertility was assessed by placing 2-day-old single male of each genotype with three 4-day-old virgin *y w* females or three 4-day-old virgin females of each genotype with three 4-day-old virgin *y w* males. Flies were transferred to fresh vials every 5 days for three times. The total number of progeny produced over 20 days by each animal was counted. Data are represented as mean ±SD of 3 independent sets of experiments. For longevity analyses, newly eclosed males of each genotype were collected and housed at a density of 5 flies per vial. Flies were transferred to fresh food every 3–4 days, and dead flies were counted every day until all died. Data are represented as mean of 3 independent sets of experiments.

## Generation of NGLY1 overexpression transgenes

Human *NGLY1* cDNA in *pCMV6-AC* vector (clone SC320763, OriGene) was used as template for site-directed mutagenesis to introduce the c.1205_1207del clinical mutation (*Enns et al., 2014*), which results in the generation of NGLY1-ΔR402. Wild-type and ΔR402 cDNAs were transferred from *pCMV6-AC* to *pUAST-attB* vector by *EcoRI-XhoI* double digestion and ligation, verified by sequencing, and integrated into the *VK31* docking site by ΦC31-mediated transgenesis (*Bischof et al., 2007*; *Venken et al., 2006*).

## Gut clearance assay and visualization of the acid zone

Larvae were raised on standard food supplemented with 0.05% bromophenol blue (BPB). Wandering larvae were collected from the side of the vial with a wet paint brush, transferred to a petri dish lined with wet Whatman paper and monitored for gut clearance until puparium formation. About 30 larvae were scored for each genotype at each time point. Each data point is from three independent experiments. For larval gut acidification studies, 72 hours after egg deposition, larvae were transferred to standard food containing 0.05% BPB and dissected after 12 hours. For examination of adult acid zones, one-day old animals were fed on food supplemented with 0.05% BPB for two days and then dissected. The images were taken by ToupCam Camera and analyzed by ToupView software.

## RTL spotting assay

RTL spotting assay was carried out using *png1Δ* cells (*png1::KanMX4 Mata his3Δ1 leu2Δ0 met15Δ0 ura3Δ0)* and pRS313-$_{GAL4}$RTL essentially as described previously (*Masahara-Negishi et al., 2012*). In brief, strains harboring the RTL expression plasmid were spotted on to SC-histidine-uracil or SC-histidine-uracil-leucine medium containing 2% galactose (w/v), and plates were incubated at 30°C for 3 days. Photographs of the plates were taken using FUJIFILM LAS-3000 mini (Fujifilm Co., Tokyo, Japan).

## Cycloheximide decay assay

*png1Δ* cells harboring the pRS315-$_{GPD}$FLAG-RTAΔ (*Hosomi et al., 2010*) were grown at 30°C in SC-leucine liquid medium. Cycloheximide was added at t = 0 min (final concentration, 4 μg/ml), and the samples were collected at the indicated times and subjected to SDS-PAGE, followed by immunoblotting with anti-DYKDDDDK antibody 1:10,000 (Wako Cat# 018–22381, RRID:AB_10659453). Phosphoglycerate kinase (Pgk1) was used as a loading control and was probed with anti-Pgk1 antibody 1:10,000 (Molecular Probes Cat# A-6457, RRID:AB_221541).

## Western blotting

Proteins were extracted from whole larvae in lysis buffer containing protease inhibitor cocktail (Promega). The following antibodies were used: rabbit anti-NGLY1 1:500 (Sigma-Aldrich Cat# HPA036825, RRID:AB_10672231), mouse anti-tubulin 1:1000 (Santa Cruz Biotechnology Cat# sc-8035, RRID:AB_628408), mouse anti-actin 1:1000 (DSHB Cat# 224-236-1, RRID:AB_10571933), rabbit anti-Pngl 1:250 (*Funakoshi et al., 2010*), rabbit anti-Dpp 1:1000 (*Akiyama and Gibson, 2015*), mouse anti-HA 1:20,000 (Sigma-Aldrich Cat# B9183, RRID:AB_439706), goat anti-rabbit-HRP and goat anti-mouse-HRP 1:2000 (Jackson ImmunoResearch Laboratories). Western blots were

developed using Pierce ECL Western Blotting Substrates (Thermo Scientific). The bands were detected using an ImageQuant LAS 4000 system from GE Healthcare. At least three independent immunoblots were performed for each experiment.

### Immunostaining

The following antibodies were used: rabbit anti-pSMAD3 1:250 (Abcam Cat# ab52903, RRID:AB_882596), guinea pig anti-Labial 1: 1000 (*Guo et al., 2013*), rabbit-anti Dpp 1:100 (*Akiyama and Gibson, 2015*), mouse anti-βGAL 1:50 (DSHB Cat# 40-1a, RRID:AB_528100), mouse anti-Fas3 1:50 (DSHB Cat# 7G10 anti-Fasciclin III, RRID:AB_528238), mouse anti-GFP 1:500 (Thermo Fisher Scientific Cat# 33–2600, RRID:AB_2533111), goat anti-rabbit-Cy3 1:500, goat anti-mouse-Cy5 1:500 (Jackson ImmunoResearch Laboratories). Confocal images were taken with Leica TCS-SP8 microscope. All images were acquired using Leica LAS-SP software. Amira 5.2.2 and Adobe Photoshop CS6 were used for processing and Figures were assembled in Adobe Illustrator CS6.

## Acknowledgements

We thank Huda Zoghbi for her support; Kartik Venkatachalam and Noah Shroyer for discussions; Hugo Bellen and Kevin Lee for comments on the manuscript; The Bloomington *Drosophila* Stock Center (NIH P40OD018537), the Developmental Studies Hybridoma Bank, Benjamin Ohlstein, Takuya Akiyama and Matthew Gibson for reagents. This work was supported by the Grace Science Foundation through Texas Children's Hospital. Work in Jafar-Nejad laboratory is also supported by the NIH (R01GM084135 and R01DK109982). Imaging was performed at the Confocal Microscopy Core of the BCM IDDRC (U54HD083092; the Eunice Kennedy Shriver NICHD)

## Additional information

### Funding

| Funder | Grant reference number | Author |
|---|---|---|
| Grace Wilsey Foundation | Research Grant | Tadashi Suzuki<br>Hamed Jafar-Nejad |
| National Institutes of Health | R01GM084135<br>R01DK109982 | Hamed Jafar-Nejad |

The funders had no role in study design, data collection and interpretation, or the decision to submit the work for publication.

### Author contributions

Antonio Galeone, Conceptualization, Investigation, Methodology, Writing—original draft, Writing—review and editing; Seung Yeop Han, Conceptualization, Investigation, Writing—review and editing; Chengcheng Huang, Investigation, Writing—review and editing; Akira Hosomi, Investigation, Methodology, Writing—review and editing; Tadashi Suzuki, Conceptualization, Supervision, Funding acquisition, Methodology, Writing—review and editing; Hamed Jafar-Nejad, Conceptualization, Supervision, Funding acquisition, Methodology, Writing—original draft, Writing—review and editing

### Author ORCIDs

Antonio Galeone  http://orcid.org/0000-0002-0201-3089
Hamed Jafar-Nejad  http://orcid.org/0000-0001-6403-3379

### Decision letter and Author response

Decision letter https://doi.org/10.7554/eLife.27612.021
Author response https://doi.org/10.7554/eLife.27612.022

## Additional files

**Supplementary files**
• Transparent reporting form
DOI: https://doi.org/10.7554/eLife.27612.019

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
