## [Decision Letter]

Thank you for submitting your article "Tissue-specific regulation of BMP signaling by *Drosophila* N-glycanase 1" for consideration by *eLife*. Your article has been reviewed by two peer reviewers, and the evaluation has been overseen by K VijayRaghavan as the Senior Editor and Reviewing Editor. The reviewers have opted to remain anonymous.

The reviewers have discussed the reviews with one another and the Reviewing Editor has drafted this decision to help you prepare a revised submission.

Summary:

In this manuscript, the authors investigate the role of the *pngl* gene (which encodes an enzyme that removes N-linked glycans) during *Drosophila* development. This study is highly relevant as mutations in the human ortholog of this gene (NGLY1) are responsible for a congenital disorder of glycosylation where patients present with major developmental delays. Using genetic deficiencies, deletions and RNAi interference, the authors identify a role for this gene in Dpp signaling during early development that results in phenotypic abnormalities in larvae. Specifically, they demonstrate that loss of *pngl* influences the extent of Dpp propagation in PS3 and PS7 of embryonic development, thereby affecting mesoderm to endoderm signaling. This study is thorough, well-controlled and will provide important insights into signaling pathways and cellular mechanisms that may be responsible for the phenotypes observed in human NGLY1 patients.

This study does represent an important step in understanding how the loss of this gene influences a particular signaling pathway during a stage of development and could be very useful in beginning to interpret some of the developmental defects occurring in the patients. It would be very informative to know if (and how) the glycosylation status of *dpp* is altered upon loss of *pngl,* but we do see that is beyond the scope of the current study.

Please use the specific comments from the reviewers, some overlapping, to submit a revised manuscript.

Essential revisions:

*Comments of Reviewer #1:*

The overall concerns with the manuscript at this point are some of the interpretations and lack of biochemical mechanistic insight into the link between N-glycanase activity and Dpp homodimerization.

1) Why are *Pngl* mutants lethal? Mutants that lead to loss of the Copper cell region do not result in lethality (Cell Tissue Res. 2001 Oct: 306(1): 167-78). *how-Gal4* driven knockdown also results in lethality but does affect gastric caeca formation. This suggests that lethality is not due to loss of the caeca.

2) *Mef2-Gal4* and *how-Gal4* expression of human NGLY1 partially rescues *Pngl* lethality. *Mef2-Gal4* and *how-Gal4* expression of *Pngl* RNAi results in 0 and 28% survivability and *Mef2-Gal4* and *how-Gal4* expression of *dpp* RNAi results in loss of the acid region and caeca and 8 and 80% survivability, leading the authors to conclude that *Pngl* plays a *dpp* independent role. This may indeed be the case and should be mentioned in the Abstract. However, an alternative explanation is that *dpp* is required in other tissues and the knockdown by these drivers is not effective in the other tissues.

3) Subsection “Loss of BMP signaling in *Pngl^–/–^* embryonic endoderm is caused by impaired BMP autoactivation in VM”: *Mef>tkv^CA^* rescues *Pngl* lethality. Given the concerns that lethality is not due to caeca or acid region loss, this would argue that *Pngl* is required for BMP/ (maybe TGF-B) signaling elsewhere.

4) Do *Mef>tkv^CA^ Pngl* minus flies that make it to adulthood have a Copper cell region. It would be interesting to know if *Pngl* is required for proper midgut formation during metamorphosis.

5) Mutants that make it to adulthood are sterile. Do these animals have ovaries/testis that are consistent with loss of *dpp*, i.e. germline stem cell loss? Knowing this might help to establish *Pngl* as a gene required for a subset of BMP signaling pathway events. It would especially be interesting as *gbb* is also required for germline stem cell maintenance.

6) From the Abstract, the authors write,"[…]without affecting BMP signaling in other contexts[…]" It's impossible to be sure it is not affecting BMP signaling in other contexts. It's just that the authors were not able to identify any others. The same concern relates to the last sentence of the subsection “BMP signaling from VM to endoderm is impaired upon loss of *Pngl*”. The word "specifically" should be removed.

7) Do *Pngl* genes harboring a C303A mutation fail to rescue the *Pngl* mutant fly phenotypes. This would at least genetically link the catalytic domain/activity to described phenotypes. There is no data in this paper to how *Pngl* affects *dpp*.

8) Why did the authors, in Figure 6, do Western Blots on whole larvae and not midguts versus carcass (midgut minus)? Changes in dimmer accumulation in carcass could help identify other tissues where *Pngl* may act.

9) "BMP ligands can signal both as homodimers and as heterodimers (Bragdon et al., 2011; 383 O'Connor et al., 2006)."

If so, have the authors examined the tissues where the ligands act as homodimers? Do they have phenotypes? Changes in accumulation on Western Blots?

*Comments of Reviewer #2:*

1) As the mesodermal knockdown of *pngl* was more severe that the knockdown of *dpp*, can the authors comment on reasons for this (are there other suspected targets of *Pngl* in the mesoderm)?

2) Clarification with regard to rescue experiments-is Dpp-GFP processed and dimerized in the same manner as endogenous Dpp? Does it have the same N-glycans? If not, could this affect your interpretation (or perhaps give clues as to what region of Dpp is affected by *Pngl*)?

3) In *pngl* mutants, can you distinguish between Dpp protein simply being produced at a lower level (due to instability, higher turnover, etc.) vs. Dpp protein not being able to mediate autoregulation? Or are these different ways of saying the same thing?

4) Can the authors comment on whether any of the major phenotypes seen in the NGLY1 patients are thought to be due to changes in BMP signaling?

---

## [Author Response]

*Essential revisions:*

*Comments of Reviewer #1:*

The overall concerns with the manuscript at this point are some of the interpretations and lack of biochemical mechanistic insight into the link between N-glycanase activity and Dpp homodimerization.1) Why are Pngl mutants lethal? Mutants that lead to loss of the Copper cell region do not result in lethality (Cell Tissue Res. 2001 Oct: 306(1): 167-78). how-Gal4 driven knockdown also results in lethality but does affect gastric caeca formation. This suggests that lethality is not due to loss of the caeca.

We thank the reviewer for raising this important point. Mutants with loss of acid zone usually do not show lethality. Moreover, our data indicate that *dpp* knockdown by *how-GAL4*, which is expressed after gastric caeca anlagen is formed, does not lead to significant lethality. This is not true in *how-GAL4*-mediated *Pngl* knockdown experiments, where the lethality is remarkably high although gastric caeca formation is not affected. Therefore, although we think gastric caeca shortening contributes to *Pngl* lethality (please see below), these data clearly suggest that other events are involved in the lethality of *Pngl* mutants. At the time of the original submission, we had data showing a Dpp-independent midgut phenotype (food accumulation) in *Pngl* mutant and mesoderm-specific knockdown larvae. However, we decided not to include the data in the first version because we do not know the molecular mechanism for this phenotype, and the manuscript was focused on BMP signaling. Since both reviewers have inquired about the reason behind the discrepancy between *dpp* knockdown and *Pngl* mutant/knockdown phenotypes, we now include the data in the revised Figure 4. *Pngl*-mutant wandering larvae accumulate food in the midgut and are not able to properly empty their midgut before puparium formation. Comparing knockdown data for *dpp* and *Pngl* by pan-mesodermal drivers reveals that the food accumulation phenotype in *Pngl* larvae is at least in part and perhaps fully independent of impairment in BMP signaling. Specifically, *how>dpp^RNAi^* larvae do not show any food accumulation phenotype, but *how>Pngl^RNAi^* larvae show a rather severe food accumulation phenotype. It is noteworthy that *Mef2>dpp^RNAi^* larvae show some degree of food accumulation. This phenotype is milder than that observed in *Mef2>Pngl^RNAi^* larvae and *Pngl* mutants, and might be due to defects caused by complete loss of Dpp in the mesoderm, independently of the function of *Pngl*. Nevertheless, in the manuscript we call the food accumulation phenotype partially independent from Dpp signaling to be on the safe side.

By studying several alleles of *dpp* which specifically affect its expression in PS3 and PS7 in the embryonic visceral mesoderm, the Hoffmann group showed that loss of gastric caeca is accompanied by developmental delay and some degree of larval and pupal lethality (Masucci and Hoffmann, 1993). In agreement with this report, we believe our data indicate that shortening of gastric caeca contributes to the lethality of *Pngl* mutants. As shown in the original Figure 5 (revised Figure 6), overexpression of TkvCA restores BMP signaling in *Pngl* mutant embryonic midgut and rescues the lethality of *Pngl* mutants around 30%. Moreover, the rescue of *Pngl* mutant lethality by *how>NGLY1* is weaker than the rescue with *Mef2>NGLY1* (revised Figure 3). Lastly, ~30% of *how>Pngl^RNAi^*animals reach adulthood, even though they lose the acid zone and show food accumulation. We conclude that the lethality of *Pngl* mutants arises from at least two sources: to some extent from BMP-dependent defects in the midgut (mostly due to the gastric caeca shortening), to a larger extent from BMP-independent defects in the mesoderm, and potentially due to minor contributions from non-mesodermal tissues. The importance of the structural and functional integrity of visceral mesoderm in *Drosophila* larvae is an understudied area. Among the small number of studies that we found discussing the importance of visceral mesoderm, one of them seems to show a correlation between a failure to empty the gut due to functional defects in visceral mesoderm and lethality (Bland et al., 2010). Unfortunately, our current data do not allow us to draw strong conclusions on the extent to which the food accumulation phenotype contributes to the lethality of *Pngl* larvae. We have used the following sentences in the revised Discussion to convey this message: “Phenotypic analysis of *Pngl* mutants combined with rescue and KD experiments suggest that a failure to properly empty the gut before puparium formation contributes to lethality in these animals. The molecular mechanisms for the food accumulation phenotype and other potential *Pngl^–/–^*phenotypes contributing to lethality are still under investigation.”

2) Mef2-Gal4 and how-Gal4 expression of human NGLY1 partially rescues Pngl lethality. Mef2-Gal4 and how-Gal4 expression of Pngl RNAi results in 0 and 28% survivability and Mef2-Gal4 and how-Gal4 expression of dpp RNAi results in loss of the acid region and caeca and 8 and 80% survivability, leading the authors to conclude that Pngl plays a dpp independent role. This may indeed be the case and should be mentioned in the Abstract. However, an alternative explanation is that dpp is required in other tissues and the knockdown by these drivers is not effective in the other tissues.

In the revised version, we mention the issue raised by the reviewer in the Abstract as well as in Results and Discussion. As discussed above, we now present evidence for a food accumulation phenotype in *Pngl* mutant and mesodermal KD animals that is at least partially Dpp-independent. Given that overexpression of NGLY1 in the mesoderm of *Pngl* mutants results in the formation of adults at greater than 80% of the Mendelian ratio, *Pngl* seems to be primarily required in the mesoderm during fly development. Having said that, the reviewer’s point is well taken: our data are not sufficient to rule out a role for *Pngl* in non-mesodermal tissues. Accordingly, we have added the following phrase to the last paragraph of the Discussion: “Although we cannot exclude that *Pngl* might play important roles in other cell types as well, our data indicate that *Pngl* is primarily required in the mesoderm during midgut development[…]”

3) Subsection “Loss of BMP signaling in Pngl^–/–^ embryonic endoderm is caused by impaired BMP autoactivation in VM”: Mef>tkv^CA^ rescues Pngl lethality. Given the concerns that lethality is not due to caeca or acid region loss, this would argue that Pngl is required for BMP/ (maybe TGF-B) signaling elsewhere.

As explained above, our data indicate that gastric caeca shortening most likely contributes to lethality, in agreement with a previous report (Masucci and Hoffmann, 1993). Therefore, we believe that the partial rescue of *Pngl* lethality by *Mef2*>tkvCA can be explained by the rescue of gastric caeca. However, we agree with the reviewer that it is possible that *Pngl* is required in other contexts for TGF-β or BMP signaling during embryonic or larval development. Accordingly, we have modified the Results section as follows: “These observations further indicate that impaired BMP signaling is partially responsible for the lethality of *Pngl^–/–^* animals. This is likely due to gastric caeca shortening (Masucci and Hoffmann, 1993), although we cannot exclude that *tkvCA* overexpression affects unknown BMP-related defects in other parts of mesoderm in these mutants.”

4) Do Mef>tkv^CA^ Pngl minus flies that make it to adulthood have a Copper cell region. It would be interesting to know if Pngl is required for proper midgut formation during metamorphosis.

To address this point, we have investigated the adult midgut of *Pngl* escapers and compared them with flies rescued by overexpressing *tkvCA* with *Mef2-GAL4.* As shown in the revised Figure 6—figure supplement 1, *tkvCA* rescues the adult Copper cell region and the midgut length in *Pngl* mutants.

5) Mutants that make it to adulthood are sterile. Do these animals have ovaries/testis that are consistent with loss of dpp, i.e. germline stem cell loss? Knowing this might help to establish Pngl as a gene required for a subset of BMP signaling pathway events. It would especially be interesting as gbb is also required for germline stem cell maintenance.

To address this point, we have tested whether loss of *Pngl* causes impairment of BMP signaling in germline stem cells (GSCs) of ovaries and testes in *Pngl* adult escapers 3-4 days after eclosion (n>10 for each). We have stained both tissues by pMad antibody to test BMP signaling and Vasa antibody to mark germline cells and their progeny. Our data suggest that BMP signaling in germaria of *Pngl* mutant ovaries is not lost, as we see pMad co-localized with Vasa in the anterior part of the germarium. Unfortunately, we cannot test GSC maintenance and BMP signaling in an age-dependent manner in mutant germaria of *Pngl* adult escapers due to their short lifespan. Notably, we have found a severe decrease in pMad expression in the anterior part of the mutant testis in male *Pngl* adult escapers, strongly suggesting BMP signaling impairment in this tissue. These data indicate that the reviewer is correct: although among the BMP-dependent processes during embryonic and larval stages that we have studies so far midgut development is the only context affected by the loss of *Pngl, Pngl* is likely to have important roles in some additional cell types as well. We have modified the revised version to reflect this notion, as mentioned in our response to questions 3 and 4. However, we feel that it’s premature to add the testis and ovary data to the manuscript, as both tissues also show morphological abnormalities that are not likely to be caused by the loss of BMP signaling. We will need to perform additional in-depth studies on these tissues to bring them to the same level of scrutiny applied in our midgut studies, and the manuscript is already somewhat long, especially after the addition of food accumulation data. We hope that the editor and reviewer will agree with our strategy of not showing the ovary/testis data but clearly mentioning that specific cell types other than visceral mesoderm are likely to require the function of *Pngl* as well.

6) From the Abstract, the authors write,"[…]without affecting BMP signaling in other contexts[…]" It's impossible to be sure it is not affecting BMP signaling in other contexts. It's just that the authors were not able to identify any others. The same concern relates to the last sentence of the subsection “BMP signaling from VM to endoderm is impaired upon loss of Pngl”. The word "specifically" should be removed.

We agree with the reviewer and have modified the text to address this issue in the Abstract and the Results section and removed “specifically”.

7) Do Pngl genes harboring a C303A mutation fail to rescue the Pngl mutant fly phenotypes. This would at least genetically link the catalytic domain/activity to described phenotypes. There is no data in this paper to how Pngl affects dpp.

This was a great suggestion (like a number of other points raised by the reviewer). As shown in new panels in the revised Figure 3), transgenic overexpression of *Pngl* harboring a single mutation in its catalytic domain (C303A) in the mesoderm of *Pngl* mutants failed to rescue the gastric caeca length and the acid zone, even though Western blots show that this mutation does not affect the expression level of Pngl. In contrast, overexpression of wild-type Pngl by the same driver (*Mef2*-GAL4) fully rescued these phenotypes. These data indicate that *Pngl*’s deglycosylation activity is essential for the regulation of BMP pathway in larval midgut.

8) Why did the authors, in Figure 6, do Western Blots on whole larvae and not midguts versus carcass (midgut minus)? Changes in dimmer accumulation in carcass could help identify other tissues where Pngl may act.

To address this issue, we have performed carcass vs. midgut Western Blot in a non-reducing condition and probed with Dpp antibody. As shown in Figure 7—figure supplement 3, the level of Dpp dimer is decreased not only in the midgut but also in other parts of the larval body. This could mean that either *Pngl* regulates BMP signaling by affecting Dpp dimer levels in other larval tissues, or that Dpp dimers are only important in the midgut and although they are decreased elsewhere, Dpp-Gbb heterodimers compensate for the lack of Dpp dimers in most other tissues. We have discussed both of these possibilities in the revised version (Discussion).

9) "BMP ligands can signal both as homodimers and as heterodimers (Bragdon et al., 2011; 383 O'Connor et al., 2006)."If so, have the authors examined the tissues where the ligands act as homodimers? Do they have phenotypes? Changes in accumulation on Western Blots?

Cell-based assays have indicated that BMP ligands can signal both as homodimers and heterodimers. However, the relative in vivo contribution of BMP heterodimer versus homodimer in most tissues is not well known. Therefore, unfortunately we do not have the knowledge required to address the reviewer’s comment.

*Comments of Reviewer #2:*

1) As the mesodermal knockdown of pngl was more severe that the knockdown of dpp, can the authors comment on reasons for this (are there other suspected targets of Pngl in the mesoderm)?

We thank the reviewer for raising this point. As described in our response to questions 1-3 of reviewer 1, we have added a new figure (revised Figure 4) to address this issue. Unfortunately, we do not know the target(s) of *Pngl* involved in this phenotype, as it seems to be mostly BMP-independent.

2) Clarification with regard to rescue experiments-is Dpp-GFP processed and dimerized in the same manner as endogenous Dpp? Does it have the same N-glycans? If not, could this affect your interpretation (or perhaps give clues as to what region of Dpp is affected by Pngl)?

To clarify this point, we overexpressed *dpp-GFP* by *Mef2-GAL4* and tested the Dpp pattern by anti-GFP Western blotting in reducing and non-reducing conditions. As previously reported in S2 cells (Teleman and Cohen, 2000), we found that the precursor and mature forms of Dpp-GFP match the expected size and pattern in reducing conditions (Author response Figure 1, left lane). In addition, in non-reducing conditions the size of the observed bands corresponds to Dpp-GFP dimers, and they show the same pattern as the endogenous Dpp (Author response Figure 1, right lane). These data suggest that the GFP polypeptide which has been inserted in Dpp protein does not affect its processing and dimerization. Moreover, we have checked the GFP sequence inserted in Dpp by consulting the Experimental Procedures of the paper which reported the generation of this transgene (Teleman and Cohen, 2000) and looked for potential *N*- glycosylation consensus sequences (N-X-S/T). None of the 13 asparagine residues present in this fragment conforms to the *N*-glycosylation consensus sequence. Since the GFP has been inserted in the context of the full length Dpp (without any deletions in the Dpp protein), the Dpp-GFP protein most likely possesses the same *N*-glycans as the endogenous Dpp.

**Author response image 1. respfig1:** Dpp-GFP shows dimerization and processing patterns similar to Dpp. Larval extracts from *Mef2>dpp-GFP*were run under reducing (left lane) and non-reducing (right lane) conditions and probed with an anti-GFP antibody.

3) In pngl mutants, can you distinguish between Dpp protein simply being produced at a lower level (due to instability, higher turnover, etc.) vs. Dpp protein not being able to mediate autoregulation? Or are these different ways of saying the same thing?

Four observations in the original manuscript were used to conclude that Dpp autoregulation in the visceral mesoderm is impaired: presence of initial, Ubx-dependent expression of *dpp* in stage 13 *Pngl* embryos, lack of pMad in the visceral mesoderm of these embryos, rescue of pMad expression in endoderm by mesodermal overexpression of Dpp-GFP, and rescue of pMad expression in the endoderm by overexpression of TkvCA in the mesoderm. We believe that it is difficult to reconcile the hypothesis of a simple reduction in Dpp as the cause of these phenotypes with the last line of evidence mentioned here. However, the reviewer’s comment prompted us to test whether reducing the *dpp* expression in mesoderm by another strategy could impair BMP autoregulation. The *dpp^S2^* allele harbors a chromosomal breakpoint in regulatory elements that control *dpp* expression in the visceral mesoderm and results in a complete loss of *dpp* expression in the visceral mesoderm in homozygosity, without affecting its expression in other embryonic tissues (Masucci and Hoffmann, 1993). We reasoned that if decreasing Dpp production levels is sufficient to explain the *Pngl* midgut phenotypes, animals heterozygous for this allele, which will presumably lose 50% of *dpp* expression in the visceral mesoderm, should show defects in the propagation of the Dpp expression domain and expression of pMad in embryonic midgut and potentially some BMP-dependent defects in the larval midgut. As the new panels in the revised Figure 5) show, in stage 13 *dpp^S2^* heterozygous embryos, the level of Dpp protein in PS7 is even less than that Dpp level in *Pngl* homozygous embryos. However, it can lead to Dpp spreading at stage 14. Moreover, as shown in a new supplemental figure of Figure 5, *dpp^S2^* heterozygous animals do not exhibit defects in the formation of second midgut compartment and Labial expression in the embryo and in the acid zone in third instar larvae. These observations further support the conclusion that impairment of the Dpp autoregulatory loop underlies the loss of BMP signaling in *Pngl* mutant midguts.

4) Can the authors comment on whether any of the major phenotypes seen in the NGLY1 patients are thought to be due to changes in BMP signaling?

We have added some sentences to the last paragraph of the Discussion section to address this issue.